

# An empirical spatiotemporal description of the global surface-atmosphere carbon fluxes: opportunities and data limitations

Jakob Zscheischler[1,2], Miguel D. Mahecha[2,3,4], Valerio Avitabile[5], Leonardo Calle[6], Nuno Carvalhais[2,7], Philippe Ciais[8], Fabian Gans[2], Nicolas Gruber[9], Jens Hartmann[10], Martin Herold[5], Kazuhito Ichii[11,12], Martin Jung[2], Peter Landschützer[9,13], Goulven. G. Laruelle[14], Ronny Lauerwald[14,15], Dario Papale[16], Philippe Peylin[7], Benjamin Poulter[6,17], Deepak Ray[18], Pierre Regnier[14], Christian Rödenbeck[1], Rosa M. Roman-Cuesta[5], Christopher Schwalm[19], Gianluca Tramontana[16], Alexandra Y. Tyukavina[20], Riccardo Valentini[21], Guido van der Werf[22], Tristram O. West[23], Julie E. Wolf[23], and Markus Reichstein[2,3,4]

[1]Institute for Atmospheric and Climate Science, ETH Zurich, Universitätstr. 16, 8092 Zurich, Switzerland

[2]Max Planck Institute for Biogeochemistry, Hans-Knöll-Str. 10, 07745 Jena, Germany

[3]German Centre for Integrative Biodiversity Research (iDiv), Deutscher Platz 5e, 04103 Leipzig, Germany

[4]Michael Stifel Center Jena for Data-Driven and Simulation Science, 07743 Jena, Germany

[5]Wageningen University & Research, Laboratory of Geo-Information Science and Remote Sensing, Droevendaalsesteeg 3, 6708 PB Wageningen, the Netherlands

[6]Institute on Ecosystems and Department of Ecology, Montana State University, Bozeman, MT 59717, USA

[7]CENSE, Departamento de Ciências e Engenharia do Ambiente, Faculdade de Ciências e Tecnologia, Universidade NOVA de Lisboa, Caparica, Portugal

[8]Laboratoire des Sciences du Climat et de l'Environnement, CEA-CNRS-UVSQ, F-91191, Gif sur Yvette, France

[9]Institute of Biogeochemistry and Pollutant Dynamics, ETH Zurich, Zurich, Switzerland

[10]Institute for Geology, CEN- Center for Earth System Research and Sustainability, University of Hamburg, Hamburg, Germany 55, D-20146 Hamburg, Germany

[11]Department of Environmental Geochemical Cycle Research, Agency for Marine-Earth Science and Technology, Yokohama, Japan.

[12]Center for Global Environmental Research, National Institute for Environmental Studies, Tsukuba, Japan.

[13]Max Planck Institute for Meteorology, Bundesstr. 53, Hamburg, Germany

[14]Dept. Geoscience, Environment & Society (DGES), CP160/02, Université Libre de Bruxelles, 1050 Bruxelles, Belgium

[15]College of Engineering, Mathematics and Physical Sciences, University of Exeter, EX4 4QE Exeter, Devon, UK

[16]Department for Innovation in Biological, Agro-food and Forest systems (DIBAF), University of Tuscia, Viterbo, 01100, Italy





[17]NASA Goddard Space Flight Center, Biospheric Sciences Laboratory, Greenbelt, MD 20771, USA

[18]Institute on the Environment, University of Minnesota Twin-Cities, USA

[19]Woods Hole Research Center, Falmouth MA 02540, USA

[20]Department of Geographical Sciences, University of Maryland, College Park, MD, USA

[21]CMCC, Via A. Imperatore, 16, 73100, Lecce, Italy

[22]Faculty of Earth and Life Sciences, Vrije Universiteit Amsterdam, the Netherlands

[23]Joint Global Change Research Institute, Pacific Northwest National Laboratory, College Park, MD, USA

*Correspondence to*: J. Zscheischler (jakob.zscheischler@env.ethz.ch) and M. D. Mahecha (mmahecha@bgc-jena.mpg.de).

**Abstract.** Understanding the global carbon (C) cycle is of crucial importance to map current and future climate dynamics
relative to global environmental change. A full characterization of C cycling requires detailed information on spatiotemporal
patterns of surface-atmosphere fluxes. However, relevant C cycle observations are highly variable in their coverage and
reporting standards. Especially problematic is the lack of integration of vertical oceanic, inland freshwaters and terrestrial
carbon dioxide ($CO_2$) exchange. Here we adopt a data-driven approach to synthesize a wide range of observation-based
spatially explicit surface-atmosphere $CO_2$ fluxes from 2001 and 2010, to identify the state of today's observational
opportunities and data limitation. The considered fluxes include vertical net exchange of open oceans, continental shelves,
estuaries, rivers, and lakes, as well as $CO_2$ fluxes related to gross primary productivity, terrestrial ecosystem respiration,
fire emissions, loss of tropical aboveground C, harvested wood and crops, as well as fossil fuel and cement emissions. Spatially
explicit $CO_2$ fluxes are obtained through geostatistical and/or remote sensing-based upscaling; minimizing biophysical or
biogeochemical assumptions encoded in process-based models. We estimate a global bottom-up net C exchange (NCE)
between the surface (land, ocean, and coastal areas) and the atmosphere. Uncertainties for NCE and its components are
derived using resampling. In most continental regions our NCE estimates agree well with independent estimates from other
sources. This holds for Europe (mean±1 SD: 0.80±0.16 PgC/yr, positive numbers are sources to the atmosphere), Russia (-
0.02±0.49 PgC/yr), East Asia (1.76±0.38 PgC/yr), South Asia (0.25±0.16 PgC/yr), and Australia (0.22±0.47 PgC/yr). Our
NCE estimates also suggest large C sink in tropical areas. The global NCE estimate is -6.07±3.38 PgC/yr. This global
bottom-up value is the opposite direction of what is expected from the atmospheric growth rate of $CO_2$, and would require an
offsetting surface C source of 4.27±0.10 PgC/yr. This mismatch highlights large knowledge and observational gaps in
tropical areas, particularly in South America, Africa, and Southeast Asia, but also in North America. Our uncertainty
assessment provides the basis for designing new observation campaigns. In particular, we lack seasonal monitoring of shelf,
estuary and inland water-atmosphere C exchange. Also, extensive $pCO_2$ measurements are missing in the Southern Ocean.
Most importantly, tropical land C fluxes suffer from a lack of in-situ observations. The consistent derivation of data
uncertainties could serve as prior knowledge in multi-criteria optimization such as the Carbon Cycle Data Assimilation
System (CCDAS) without overstating data credibility. Furthermore, the spatially explicit flux estimates may be used as a
starting point to assess the validity of countries' claims of reducing net C emissions in climate change negotiations.



## 1 Introduction

The global carbon (C) cycle is crucial for sustaining life on Earth (Vernadsky, 1926). Humans have largely modified the C cycle over centuries if not millennia (Pongratz et al., 2009; Ruddiman, 2003). In recent times this has been laregly driven by

burning fossil fuels and rapid changes in land use. As anthropogenic C emissions are partly taken up by oceans and terrestrial vegetation, the different components of the global C cycle and the fluxes between them change over time (Houghton, 2007). A precise knowledge of the various stocks and fluxes in the C cycle is a prerequisite to monitor these changes and make well-informed predictions under future climate change.

The Global Carbon Project (GCP) has made major efforts in this direction and its annual updates of the global C budget have

become a crucial source of information for the scientific community and policy makers (Le Quéré et al., 2015). The GCP annual C budget quantifies the partitioning of anthropogenic C emissions among the atmosphere, land, and ocean components of the global C cycle, and separates the net land flux into land use change emissions and a so called 'residual land carbon sink' over non land-use affected ecosystems. The budget of the GCP focuses on annual values integrated at the global scale. An important caveat is that the GCP budget focuses solely on anthropogenic $CO_2$, i.e., it provides information

about the fate of anthropogenic emissions in natural reservoirs only while ignoring background $CO_2$ fluxes over land and ocean, some of them related to riverine C transport (Ciais et al., 2013; Regnier et al., 2013). According to the GCP, about 44% of the anthropogenic $CO_2$ emissions each years stay in the atmosphere, while the rest is taken up by the oceans (26%) and land (30%) (Le Quéré et al., 2015).

Recently, a case has been made for a globally policy-relevant integrated carbon observation and analysis system (Ciais et al.,

2014). This system would go beyond the update of global budgets, for which the $CO_2$ growth rate accurately measured at a single station (e.g. Mauna Loa) is sufficient to constrain the global multi-year time-space integral of all $CO_2$ sources and sinks. It proposes to quantify regional $CO_2$ fluxes at a relevant spatial scale to monitor the effectiveness of $CO_2$ mitigation and to detect and monitor trends of $CO_2$ losses and gains by land and terrestrial systems. This is partly relevant for monitoring country-level Intended Nationally Determined Contributions (INDCs) to keep global warming well below 2

degrees Celsius (UNFCCC, 2015). In such a system, an uncertainty assessment for each data stream is important to, for instance, identify significant regional to global trends (Ciais et al., 2014).

The steadily increasing number of Earth observations, in particular since the start of the satellite era, has improved our knowledge about the Earth system (Berger et al., 2012; Tatem et al., 2008). Especially C cycle science has benefited from globally available satellite observations and community efforts to unify in-situ observational networks such as FLUXNET on

land (Baldocchi, 2014) and Surface Ocean $CO_2$ Atlas (SOCAT) over water (Bakker et al., 2014). Combining these now abundantly available point measurements of either $CO_2$ fluxes (e.g., from eddy-covariance towers on land), or variables that can be directly related to $CO_2$ fluxes (e.g., pCO$_2$ over aquatic surfaces) with climate and remotely sensed variables (e.g.,




vegetation greenness), provides a basis to robustly upscale surface-atmosphere $CO_2$ exchange to larger areas using statistical models (Jung et al., 2011; Rödenbeck et al., 2014).

In this study we aim to characterize the maturity of current C cycle observations on ground for quantifying a spatiotemporally explicit picture of the net $CO_2$ exchange between the Earth's surface (terrestrial and aquatic) and the

atmosphere. Unlike the GCP global budget of anthropogenic $CO_2$, we consider here both natural background and anthropogenic surface-atmosphere $CO_2$ fluxes. Here, anthropogenic fluxes consist of those fluxes that occur as the result of past and present human activities since an arbitrary time in the past, such as fossil and land use emissions, climate change, reactive nitrogen creation, and ecosystem management (see Gasser and Ciais, 2013 for equations). We focus our analysis on fluxes that can be directly derived from observations. That is, we use data-driven empirical models instead of process-based

models, which are only indirectly constrained by observations. Further, we only consider 'bottom-up' estimates derived from measurements at the Earth's surface or from satellites. Inversions, which largely rely on atmospheric measurements in combination with a transport model, are not directly included but used for comparison. The goal of this analysis is to test the upscaling of local flux-related observations to regional and global budgets, and point out the limitations of the current observational networks and data-driven models used to interpolate them, for quantifying the most important $CO_2$ fluxes

exchanged between the Earth's surface and the atmosphere.

One of the major innovations of this study is combining data-driven estimates of oceanic, inland waters and terrestrial ecosystems $CO_2$ exchange and providing spatially explicit maps of the carbon exchange between the surface and the atmosphere at a monthly scale for the decade 2001-2010. At the same time, by adding emissions from fossil fuels and cement production and comparing with the annual growth rate of $CO_2$, we identify the limits of a purely data-driven C

budget. We characterize regions in which surface-atmosphere $CO_2$ fluxes are most uncertain based on the currently available data and the models used for upscaling, and thus point out regions where either more observations or a better understanding of the processes are necessary. It is not the primary goal of this study the provide the best global $CO_2$ flux inventory, but rather to identify the key uncertainties and observational shortcomings that would need to be addressed in future measurement campaigns or expansions of in-situ observatories.

The paper is structured as follows. In Sect. 2 we introduce the different data streams used in the analysis, including spatially explicit estimates of aquatic and terrestrial $CO_2$ exchange. In Sect. 3 we present the resulting combined synthesis as global maps, regionally aggregated fluxes, absolute and relative uncertainties, latitudinal averages and seasonal cycles. Sect. 4 addresses the benefits and limits of the current observational system for constraining global net $CO_2$ fluxes. Sect. 5 provides an outlook on future requirements to achieve better observationally-based net $CO_2$ flux estimates and discusses the necessity

for more consistent uncertainty estimates.



## 2    Data and Methods

We collected ensembles of data-driven estimates of the net $CO_2$ exchange between the Earth's surface and the atmosphere for the major subsystems of the Earth from 2001-2010 (Table 1). Each dataset was resampled to 1 x 1 degree spatial resolution and monthly time step; a compromise that minimizes aggregation/spatialization across different data products. In

this synthesis, we include net $CO_2$ exchange from open oceans, continental shelves, estuaries, rivers, lakes, and the land surface, which we combine with estimates of fossil fuel and cement emissions (FF). The land surface component accounts for fire emissions (Fire), loss of tropical above-ground biomass assumed to be released as $CO_2$ to the atmosphere ($E_{LUC}$), emissions of the $CO_2$ contained in harvested wood (Wood) and crops (Crops), and terrestrial ecosystem fluxes which are divided into Gross Primary Production (GPP) and Terrestrial Ecosystem Respiration (TER). We combine fluxes from

oceans, shelves, and estuaries into a homogeneous marine flux product in order to account for overlapping or missing regions from the different aquatic products (Marine, Sect. 2.2.6). We further compare the net $CO_2$ exchange derived from the combination of all the above products with the growth rate of atmospheric $CO_2$ (CGR). Data scarcity precludes including all known vertical $CO_2$ fluxes in this study. Missing fluxes include geological $CO_2$ fluxes, erosion related fluxes, non-$CO_2$ fluxes, wood product pools decay, and biofuel burning. Combining all fluxes, the overall net $CO_2$ exchange (NCE) between

the Earth's surface and the atmosphere is given as:

$$NCE = Marine + Lakes + Rivers - GPP + TER + Crops + Wood + E_{LUC} + Fire + FF. \qquad (1)$$

All units are transformed into fluxes of C per unit time. If all $CO_2$ fluxes are included, NCE should equal CGR. In contrast, negative NCE indicates uptake of $CO_2$ by the Earth's surface. All data used in this study are listed in Table 1 and for convenience available from the GEOCARBON website http://www.bgc-jena.mpg.de/geodb/geocarbon/Home.php; direct

access       and       citation       of       the       data       as       pre-processed       here       is       possible       via https://dx.doi.org/10.17871/GEOCARBON_synth_obs_v1.

### 2.1    Uncertainty estimation and propagation

For each flux term in Eq. (1) we computed mean fluxes over all available realizations of a given product, uncertainty (defined as one standard deviation (SD) of the annual mean across all realizations), interannual variability (IAV, defined

here as one SD of annual means across all available years) and the coefficient of variation (CV = IAV/mean). For NCE estimates, we randomly combined all datasets, using a single realization of each flux, to generate an estimate of NCE. This is repeated 200 times to construct an NCE ensemble, which was then used to calculate mean, SD, IAV, and CV of NCE. To estimate how the dependence between datasets affects our error estimate, for NCE we additionally estimate uncertainty by quadrature error accumulation assuming independent errors, defined classically by





$$e_{NCE} = \sqrt{\sum_{i=1}^{n} e_i^2} , \qquad (2)$$

where $e_i$ is the uncertainty for flux $i$, and $e_{NCE}$ is the uncertainty of NCE. Because this error estimate assumes independence of errors between all data sets, it is expected to result in higher uncertainties, as for instance GPP and TER have correlated errors due to the method used to separate net ecosystem production (NEP) into GPP and TER (as NEP=GPP-TER, Schulze, 2006).

## 2.2 Aquatic fluxes

### 2.2.1 Oceans

For the global open ocean flux estimate we used two complementary data-driven estimates (Table 1). Both approaches computed maps of the sea surface partial pressure of $CO_2$ ($pCO_2$). They relied on the surface ocean $CO_2$ observations from the SOCATv2 database (Bakker et al., 2014) and filled data gaps by either establishing relationships between auxiliary driver data and observations, which can then be applied to extrapolate $pCO_2$ in regions without data coverage (SOM-FFN, Landschützer et al., 2014), or by assimilating the available observations in a mass-balance model of the mixed layer and directly interpolating data gaps (Jena CarboScope mixed-layer scheme oc_v1.2, Rödenbeck et al., 2014). To test the established predictor-target relationship, the SOM-FFN method holds back a certain fraction of the observations proportional to the methods degrees of freedom for internal validation. Repeating this relationship building process and withholding different sets of validation data has created the 5 ensemble members schused for this study. For the Jena CarboScope mixed-layer scheme, we used the 5 sensitivity cases with changes in correlation length etc. as described in Rödenbeck et al (2014).

The $pCO_2$ fields of both methods have been validated against independent observations (Landschützer et al., 2014; Landschützer et al., 2015; Rödenbeck et al., 2014) and were compared with other complementary data based interpolation methods (Rödenbeck et al., 2015), illustrating their good performance in reconstructing interannual variation.

Both methods calculate the air-sea flux using a bulk formulation of the air-sea $CO_2$ transfer, driven by the air-sea $pCO_2$ difference ($\Delta pCO_2$) (Jähne et al., 1987) and a quadratic dependence of the wind speed at a height of 10 meters (Wanninkhof, 1992, Sweeney et al., 2007) updating the gas transfer coefficient to fit a mean transfer velocity of 16.5 cm per hour following Nägler (2009). High-resolution wind speeds at 10 meters are calculated from the u and v wind components of the ERA-interim wind speed analysis (Dee et al., 2011).

### 2.2.2 Shelves

For continental shelf seas we derived the $\Delta pCO_2$ from $3 \times 10^6$ surface pCO2 measurements extracted from the SOCATv2 database (Bakker et al., 2014) and observational atmospheric $pCO_2$ data (GLOBALVIEW-CO2, 2012). The local $CO_2$ air-sea flux values were then obtained using a wind-dependent quadratic formulation parameterized as in Wanninkhof et al.





(2013) and wind speeds extracted from a cross-calibrated multiplatform (CCMP) high-resolution data product for ocean surface winds (Atlas et al., 2011). The resulting local fluxes were then integrated spatially over 150 broad coastal regions (COSCATs - COastal Segmentation and related CATchments; Laruelle et al. (2013); Meybeck et al. (2006)) using distinct integration methods depending on the data density (Laruelle et al., 2014). In addition, a temporal integration was also

performed at the monthly, seasonal or yearly time scale depending on the data coverage. These temporally and regionally averaged air-sea $CO_2$ fluxes were then disaggregated using a 1-degree resolution map excluding land areas and open ocean waters using the shelf break as outer limit (Laruelle et al., 2014).

### 2.2.3    Estuaries

The $CO_2$ emissions from estuaries were derived from 161 annually averaged local $CO_2$ air-water exchange rates reported in

the literature (Laruelle et al., 2013). The data were allocated to one of the 45 coastal MARCATS regions (MARgins and CATchments Segmentation) defined in Laruelle et al. (2013) and further categorized among the 4 dominant estuarine types (i.e., small deltas, tidal systems, lagoons, fjords, see (Dürr et al., 2011)) to calculate regionally-averaged, type specific $CO_2$ emission rates. In MARCATS regions devoid of estuarine data, the global average type-dependent air-water $CO_2$ flux was used from Laruelle et al. (2013). These flux densities were then multiplied by the estuarine surface areas for each type,

estimated at 1-degree resolution from the length of the coastline and a type-specific length to estuarine surface ratio (Dürr et al., 2011).

### 2.2.4    Marine

We combined all non-inland aquatic fluxes (oceans, shelves, and estuaries) to a consistent marine product. For pixels with observations from multiple products (e.g., estuaries and oceans) we follow a "priority rule" whereby the shelves, estuaries, or

oceans observation value only (in that order) is retained. Empty pixels are gap-filled with 3 x 3 mean window. This same filter is also applied to the rest of the merged dataset to smooth out hard borders between the different estimates. This application does not significantly change the overall flux estimates, but arguably results in a more realistic interface. Note that in the merged Marine product, uncertainty and IAV could only be assessed for the ocean flux.

### 2.2.5    Rivers

Estimates of $CO_2$ evasion from streams and rivers were derived from a spatially explicit, empirical model of river water $pCO_2$ and global maps of stream surface areas and gas exchange velocities at a resolution of 0.5 degree (Lauerwald et al., 2015). The empirical $pCO_2$ model was trained on 1182 river catchments from the GLORICH database (Hartmann et al., 2014) for which robust averages of $pCO_2$ could be calculated. Steepness of terrain, terrestrial net primary production, average air temperature as well as population density were identified as predictors ($R^2=0.47$). The global maps of stream

surface area and gas exchange velocities were obtained by a GIS-based application of published empirical scaling laws (Raymond et al., 2013; Raymond et al., 2012) using topography (Lehner et al., 2008) and runoff (Fekete et al., 2002). The





CO$_2$ evasion was calculated as product of water-air pCO$_2$ gradient (assuming an atmospheric pCO$_2$ of 390 µatm), river surface areas, and gas exchange velocities. A Monte-Carlo simulation based on standard errors of the predictors in the pCO$_2$ model and uncertainty ranges for estimates of stream surface area and gas exchange velocity was run to produce 50 CO$_2$ evasion estimates.

### 2.2.6    Lakes

Estimates of CO$_2$ evasion from lakes and reservoirs were taken from Raymond et al. (2013), which reports average lake pCO$_2$, total lake/reservoir surface area, and total CO$_2$ evasion for 231 COSCAT regions (including endorheic regions). For the total lake/reservoirs surface area, data from the Global Lakes and Wetland Data base (GLWD, Lehner and Döll, 2004) were combined with an estimate for small lakes and reservoirs not represented in the GLWD using a scaling law. Here, we used the GLWD data to downscale the estimates of Raymond et al. (2013) to a continuous 1-degree resolution. For this purpose, we combined a uniform air-water CO$_2$ flux (per unit surface area) within each COSCAT region with a spatially explicit estimate of the lakes/reservoirs surface at this resolution. The small lakes/reservoirs not represented in the GLWD were assumed evenly distributed over the COSCAT area.

## 2.3    Terrestrial fluxes

### 2.3.1    GPP and TER

We used empirical, machine learning based products from FLUXCOM (www.fluxcom.org) for GPP and TER, derived from more than 200 FLUXNET sites and exclusively remote-sensing based predictor variables ("FLUCOM-RS", see Tramontana et al., 2016). The eight machine learning methods used here include artificial neural networks, four variants of model or regression tree ensembles, kernel methods (support vector machines, kernel ridge regression), and multivariate adaptive regression splines (Tramontana et al., 2016). All methods were trained on 8-daily tower based GPP and TER estimates from two NEP flux partitioning approaches (Lasslop et al., 2010; Reichstein et al., 2005) such that 16 ensemble members are available for GPP and TER (see Tramontana et al. (2016) fo details).

### 2.3.2    Crops

About 42% of global crop biomass is harvested, transported, and respired off site (Wolf et al., 2015a). The impact of this lateral C transport on fluxes can be seen at the country scale in the form of import and exports, but even more so at sub-regional scales where the movement of crop biomass to feed livestock and humans is evident (Hayes et al., 2012; West et al., 2011). To capture the spatial distribution of biogenic CO$_2$ fluxes from agricultural production, we used livestock and human emissions estimates (Wolf et al., 2015b) that are available from 2005-2011 at 0.05 degree spatial resolution. Because CO$_2$ that is taken up by crops is implicitly included in the NEP estimates from FLUXCOM, we only used CO$_2$ emissions related to livestock and human respiration. We aggregated best estimates of the data to 1 degree, added all uncertainty estimates





within one 1 degree pixel and used them as estimates for one standard deviation on the new 1 degree grid. Assuming Gaussian distributed errors we sampled 1000 values at each pixel and used 10 maps of the 5[th], 15[th],…, 95[th] quantiles as different ensemble members. Data was then linearly extrapolated back to 2001-2004. In a final step, and because it is not known in which months the emissions occur, we further distributed the annual estimates equally across all 12 months.

### 2.3.3   Wood

We use globally gridded forest harvesting data around year 2000 as described in the Supplementary Information S1. These data include fuelwood and roundwood harvested volumes in $m^3$. We translated wood volumes into units of C using a value of 0.275 MgC/$m^3$ from FAO (http://www.fao.org/docrep/w4095e/w4095e06.htm), assuming wood density of 0.55 t/$m^3$. To avoid double counting wood harvest with aboveground biomass loss in tropical areas, we use wood harvesting data only in locations where the amount of harvested wood (in C) exceed the average of $E_{LUC}$ (Sect. 2.3.4). We assume that 100% of the harvested wood is respired back to the atmosphere within a year, thus assuming no change in C stock of wood products and constant harvesting rates across years. However, C contained in harvested wood is usually emitted at a different location than where the harvest took place. We thus incorporated lateral shifts of harvested wood by redistributing wood harvest according to the consumption of wood as explained in the Supplementary Information S1 (see also Fig. S2).

### 2.3.4   $E_{LUC}$

We used two estimates for changes in C due to tropical deforestation and degradation. It is assumed here that 100% of biomass loss is converted to a $CO_2$ flux being released instantly (within a year) to the atmosphere. In reality, a fraction of tropical biomass lost decays in ecosystems (belowground biomass and slash) and a fraction is used in wood products of various lifetime. However, slash is decomposed fast and biomass from deforested areas is transformed on average to short-lived products ($\approx$ 5 years after Earles et al. (2012)).

1) Gross tropical deforestation emissions were taken from Harris et al. (2012). They represent total (above- and belowground) carbon loss from gross forest cover loss in the tropical regions due to human or natural causes (without forest recovery) for the period of 2000-2005.

2) More recent estimates of aboveground C loss in the tropics from stand-replacement disturbance of forest cover due to human or natural causes were provided by Tyukavina et al. (2015). Sample-based estimates of mean 2000-2012 aboveground C loss for each 30-m resolution forest carbon stratum were attributed to all pixels of the corresponding stratum and averaged to the 1x1 degree resolution.

We us $E_{LUC}$ only in those pixels where the average of 1) and 2) exceeds wood harvesting (Sect. 2.3.3).

### 2.3.5   Fire

We downloaded fire emissions from the Global Fire Emissions Database version 4 with small fires (GFED4s, http://www.globalfiredata.org) based on burned area from Giglio et al. (2013) and Randerson et al. (2012) and an updated





version of the biogeochemical modelling framework of van der Werf et al. (2010) to conver burned area to emissions. We included all fire types except tropical deforestation and degradation fires which are included in the aboveground biomass loss estimates (Sect. 2.3.4). For an earlier version (GFED3) a Monte Carlo simulations indicated an uncertainty of about 20% (1σ) for continental-scale estimates but these estimates turned out to be not very reliable. For example, the inclusion of small

fire burned led to an increase in burned area exceeding the previously assumed uncertainty and the current version therefore has no uncertainty assessment at pixel level. Note that GFED fire emissions depend on estimates of net primary production as computed by the CASA model. Moreover, there is currently no uncertainty estimate available at pixel-basis.

### 2.3.6 FF

We use the IER-EDGARv4.2 product for fossil fuel emissions, which was derived within the CARBONES project by the

Institute für Energiewirtschaft und Rationelle Energieanwendung (IER). It is based on the Edgar v4.2 fossil fuel spatial distribution (with the highest spatial resolution of 0.1 x 0.1 degree) and uses national consumption and global production statistics. Based on the sectorial distinguished EDGARv4.2 emissions, sector-specific and country specific temporal profiles were included (hourly, weekly and seasonal). A detailed description of the construction of the product is given at http://www.carbones.eu/wcmqs/project/ccdas/#Fossil%20Fuel.

### 15  2.4 Atmospheric growth rate

We estimate the atmospheric burden of $CO_2$ using the calculations made by the GCP (Le Quéré et al., 2015). These calculations are based on the global growth rate of atmospheric $CO_2$ (CGR) provided by the US National Oceanic and Atmospheric Administration Earth System Research Laboratory (NOAA/ESRL) and were derived from multiple stations selected from the marine boundary layer sites with well mixed background air (Ballantyne et al., 2012; Masarie and Tans,

1995).

### 2.5 Inversions

For a comparison of yearly variability, spatial patterns and latitudinal bands, we used annual means of 10 inversions collected in Peylin et al., (2013), available at the same spatial resolution after regrinding the original flux estimates and monthly temporal resolution. Not all inversions were available till 2010. Atmospheric $CO_2$ inversions estimate surface $CO_2$

fluxes such that they best fit observed atmospheric CO2 concentration gradients, using a transport model. They usually rely on prior information provided by terrestrial and oceanic biogeochemical models but are mostly independent from the bottom-up datasets included in the present synthesis (especially at continental scale). They further use FF as an input and then provide the surface-atmosphere flux excluding FF.



## 3   Results

### 3.1   Global net carbon exchange

Mean fluxes, their uncertainties, interannual variability (IAV), and CV (the mean-normalized IAV) for all individual fluxes contributing to NCE are presented in Table 2. Mean fluxes are also summarized graphically in Figure 1 (mean over 2001-

2010). Our best data driven bottom-up global estimate of NCE is -6.07±3.38 PgC / year. That means, that our data suggests a large net sink. However, the amount of C in the atmosphere is increasing by an estimated rate of 4.27±0.10 PgC / year. Combining both estimates, we obtain a C imbalance of 10.34±3.38 PgC / year (=NCE-CGR). Potential reasons for this mismatch are discussed Section 4.

Using the ensemble approach we obtain an uncertainty in NCE of ±3.38 PgC / year. With quadrature error accumulation,

taking into account the uncertainties in Marine, Rivers, GPP, TER, Crops, and $E_{LUC}$ (Eq. 1) the uncertainty of NCE is ±5.12 PgC / year. The higher uncertainty for the quadrature error accumulation is to be expected as in Eq. 1 all errors in the flux observations are assumed to be independent, whereas in fact many errors might be correlated as this clearly the case for GPP and TER. For most fluxes, uncertainty estimates strongly exceed IAV (Table 2).

### 3.2   Spatial patterns of net carbon exchange

Building a 200-member NCE ensemble enables us to provide a best estimate for a gridded average surface-atmosphere $CO_2$ flux map for the time period 2001-2010 (Figure 2a). According to these estimates, tropical land areas are a larger C sink than the mid-latitudes despite the visible forest bands in North America and Russia that function as sinks. In contrast, the high latitudes indicate a relatively small source. In the ocean, these patterns are reversed, with sources in the tropics and a sink in the mid-latitudes. Clearly, there is a strong land-sea contrast and land NCE is much higher in magnitude compared to ocean

NCE. In areas with high human population densities and active industry (Europe, Eastern China, US, South Africa), emissions from fossil fuels and cement production are clearly visible.

Absolute uncertainty of NCE generally scales with the mean flux and is highest in the most productive areas over land (Amazon basin, Congo basin, Indonesia; Figure 2b). Due to the small contribution of the oceans, absolute uncertainties are barely discernible.

Relative uncertainties however show very distinct patterns (Figure 2c). These are high on land in semi-arid and arid, and in mountainous regions (i.e., rather unproductive areas with near-zero mean) such as Australia, the Middle East, the Midwest US, the Sahel, South Africa, the Andes, and around the Tibetan Plateau. Marine C exchange is most uncertain in relative terms in the Bay of Bengal and in the Southern Ocean, which is known to be undersampled (Landschützer et al., 2014; Rödenbeck et al., 2014). In addition, linear features with high relative uncertainty are visible, especially in the Southern

Hemisphere. These are related to the borders of the clusters used for deriving homogenous regions of sea-air exchange in one of the ocean-exchange products, which are linked to strong spatial gradients in the sea surface $pCO_2$ (Landschützer et al.,





2014). Relative uncertainties are mostly below 100% for the median across latitudinal bands (Figure 2c). Only in the Southern Ocean the relative uncertainty is substantially higher, reflecting difficulties in reconstructing seasonal to interannual variabilities with weak observational constraints (Landschützer et al., 2014; Rödenbeck et al., 2014). Nevertheless, Landschützer et al. (2015) have shown that there is a better agreement between the estimates of Landschützer

et al. (2014) and Rödenbeck et al. (2014) when low frequency variability, such as decadal variability, is analysed.

Averaged over latitudinal bands, the tropics are clearly a C sink (Figure 3a), a feature of the FLUXCOM models used for GPP and TER, whereas mid-latitudes form a net C source, mostly due to fossil fuel and cement emissions surpassing natural C sinks. This latitudinal pattern is strongly driven by the terrestrial fluxes (Figure 3b). Aquatic C exchange in turn is about 5 times smaller in magnitude and shows a reversed picture with C sources in the tropics and C sinks in the extratropics (Figure

3c). The aquatic C source in the tropics is not only the result of the ocean air-sea exchange, but also of the very intense river outgassing in low latitude regions (Lauerwald et al., 2015). NCE in the mid-latitudes is strongly driven by fossil fuel emissions (blue line in Figure 3d shows NCE-FF). FF have little contribution in the tropics and the high-latitudes but turn the sink in the northern mid-latitudes into a strong C source.

We use the land cover map of 2005 from the European Space Agency (http://www.esa-landcover-cci.org/) to identify

tropical forests (all pixels where broadleaved evergreen trees dominate). Tropical forest, which covers about 2.8% of the Earth's surface, are responsible for a sink of -5.55±1.41 PgC / year, resulting in a global NCE without tropical forests of - 0.51±3.00 PgC / year. Although the best estimate is still far from the expected net global C source, which is constrained by the measured atmospheric $CO_2$ growth rate of 4.27±0.10 PgC / year, in 13% of our runs we obtain a global C source that is consistent with the atmospheric growth rate. Including missing fluxes for which we do not have spatial explicit estimates

(see Sect. 4.4) could close this gap. These considerations suggest that the C sink of tropical forests is probably strongly overestimated in our approach (FLUXCOM) and most responsible for the global mismatches.

### 3.3    Net carbon exchange over the RECCAP regions

Here we compare our NCE estimates over land with largely independent estimates of net ecosystem exchange (NEE) over continental-scale regions collected in RECCAP (REgional Carbon Cycle Assessment and Processes). These regions include

North America (NA, King et al., 2015), South America (SA, Gloor et al., 2012), Europe (EU, Luyssaert et al., 2012), Africa (AF, Valentini et al., 2014), Russia (RU, Dolman et al., 2012), East Asia (EA, Piao et al., 2012), South Asia (SAs, Patra et al., 2013), and Australia (AU, Haverd et al., 2013). No regional study is yet available for Southeast Asia (SEA). Greenland, Middle East, Ukraine, Kazakhstan and New Zealand are omitted in regional RECCAP studies because of the difficulty to obtain local ground-based observations. Ciais et al. (in revision) collected the regional estimates and combined them with

estimates of lateral transport to estimate carbon budgets for each region. NEE in Ciais et al. (in revision) minus C export by rivers should in principal be equal to our NCE estimates without FF over the same regions (Figure 4). In regions without tropical forest except NA (that is, EU, RU, EA, SAs, and AU) the estimates by Ciais et al (in revision) are within the



interquartile range of our assessment. For NA and regions containing the tropics, our approach shows a much stronger C sink.

Annual NCE-FF for all RECCAP regions amounts to -11.7±3.1 PgC / yr in contrast to -1.3±0.6 PgC / yr in Ciais et al. (in revision). If we exclude SA, AF and SEA, the numbers are -2.8±1.9 PgC / yr and -1.5±0.4 PgC / yr, respectively, bringing

both estimates in each other's uncertainty range. Hence, except for continents with tropical forest, the estimates between the two assessments match relatively well. For SA, AF and SEA, however, the two estimates even differ in sign. While our estimates indicate strong C sinks of -4.8±1.0, -2.6±1.0, and -1.5±0.4 PgC / yr, respectively, Ciais et al. (in revision) report 0.1±0.3, 0.1±0.3, and 0.0±0.2 PgC / yr.

Given that Ciais et al. (revision) rely on an independent method, this demonstrates that a good understanding of net C fluxes

exists for non-tropical areas, North America excluded. Yet it also reveals the high uncertainties in bottom-up estimates of NCE over tropical forests (see Sect. 3.2, but also Gloor et al. (2012) and Valentini et al. (2014)) and underlines the importance of long-term ground based measurement campaigns (e.g. RAINFOR, http://www.rainfor.org/, Malhi et al. (2002), and ATTO, Andreae et al. (2015), Zhou et al. (2014)).

### 3.4 Comparison with inversions

We compare NCE without FF (NCE-FF) with annual values from 10 inversions estimating the land-atmosphere $CO_2$ flux without FF (Peylin et al., 2013). While both estimates agree well in the extratropics, they show opposite patterns in the tropics (Figure 3d). The latitudinal pattern of the inversions follows a pattern similar to that of the aquatic fluxes in the present synthesis (Figure 3c), possibly related to the fact that the interiors of many continents are widely undersampled (Peylin et al., 2013), propagating the marine signal into continents. On the other hand, the estimates of GPP-TER in NCE-FF

probably have a substantial bias towards too much uptake over tropical land. The comparison suggests that C fluxes are comparably well constrained in the extratropics where bottom-up and top-down approaches agree.

The temporal evolution between both estimates show little agreement except the trend towards more net C uptake by the Earth's surface (Figure 5). Uncertainties are very high for our NCE-FF (Figure 3d). In addition, the mean annual C uptake in our estimate is about 10 PgC/yr higher than for inversions.

### 25 3.5 Monthly variability and mean seasonal cycle

NCE in the Northern hemisphere exhibits a much stronger mean seasonal cycle, ranging from a net C uptake of about 2 PgC (per month) in July to a net C release of about 0.9 PgC in December and January (Figure 6). The Southern hemisphere is always a net C sink, ranging between slightly under 1 PgC uptake in February to roughly 0.2 PgC in August and September. This illustrates the "breathing of the Earth", that is, vegetation activity largely follows the annual cycle of the sun. Northern

hemispheric NCE is strongly offset by fossil fuel emissions. The uncertainties for the Southern hemispheric seasonal cycle



are generally about 1.5-2 times lower than for the Northern hemispheric fluxes due to the larger contribution of the latter to the overall flux pattern.

## 4 Current limitations of a bottom-up spatiotemporal assessment of net carbon exchange

### 4.1 Difficulties in estimating TER and NEP over land

Correctly predicting NEP from remote sensing requires establishing universal relationships between those drivers and ecosystem respiration (TER) (Tramontana et al., 2016). However, predicting respiratory processes still poses major challenges to researchers (Trumbore, 2006). The $CO_2$ flux related to decomposition, for instance, relates to factors controlling microbial activity such as temperature, moisture availability, and the decomposable substrate material. How soil or total ecosystem respiration depends on these variables is not yet well understood. Advancing our knowledge on these processes is challenging due to both a lack of theory of respiration and the difficulty of obtaining relevant data to test models (Trumbore, 2006).

In addition to a good theory for respiration, information on disturbance history and forest age would improve the upscaling TER and NEP from sites to regions (Ciais et al., 2014). Disturbances that cause physical damage to vegetation properties tend to temporarily increase respiration and reduce photosynthesis and thus alter the balance between GPP and TER. Disturbed ecosystems are thus initially assumed to be strong C sources until plant production recovers. However, how these regrowth processes compensate a given disturbance regime cannot yet be quantified at global scales, as the area covered by disturbed ecosystems is variable and unknown (Ciais et al., 2014). For example, regrowth of vegetation after fires and other disturbances is not well sampled by FLUXNET stations. Furthermore, management can have strong effects on annual NEP of croplands which form large parts of the land surface (Jung et al., 2011). Even if all the above processes were well known and described, not all relevant drivers are available to be included in the upscaling (Tramontana et al., 2016).

If we assume that NEP is the main reason why the C budget is not closed in our approach, this raises the question why upscaled NEP has such a strong systematic bias towards a sink, particularly in the tropics (see also Jung et al., 2011). One hypothesis related to the missing information of disturbance history is that eddy-covariance towers collected in FLUXNET (which provide the empirical basis for the global data driven estimates, see Sec. 2.3.1) do not represent the different age classes of forests very well. For instance, young and regrowing forests with a generally higher-than average NEP are overrepresented in FLUXNET, i.e., the C budget of a forest depends on its age. Such an age-dependency (Amiro et al., 2010; Coursolle et al., 2012; Hyvönen et al., 2007; Magnani et al., 2007) has not been included in global upscaling of net ecosystem exchange, mostly due to missing global driver information.



### 4.2 Uncertainties in fire emissions

Fire emission estimates combine satellite-based fire data with ecosystems models. Uncertainties in global fire emission estimates are substantial and different fire products vary largely by location, vegetation type and fire weather (Ciais et al., 2014; French et al., 2011).

While GFED4 burned area estimates come with uncertainty estimates (Giglio et al., 2013), the actual uncertainty of C emissions from fires might be larger, in the order of 50% and depend regionally and temporally on the various input data sets such as burned area, small fire burned area, biomass loadings, combustion completeness, etc. Top-down assessments using for example carbon monoxide (CO) have indicated that estimates on regional scales are often not too far off but given that different top-down studies using different atmospheric models provide conflicting information on the adjustments needed to

align modelled concentrations with measured ones, this information cannot be used to provide clear uncertainty ranges

### 4.3 Seasonality for coastal and inland waters, wood and crop harvest emissions

Recently, major steps have been undertaken to resolve the spatial variability of coastal and inland water $CO_2$ fluxes (Laruelle et al., 2013; Laruelle et al., 2014; Lauerwald et al., 2015; Raymond et al., 2013). To better constrain C exchange on a monthly basis, however, the seasonal cycles of those fluxes would be necessary. For inland waters, seasonality has so far

only been investigated at regional scale (Laruelle et al., 2015; Richey et al., 2002). For shelves some seasonal estimates are currently available in temperate and high latitudes, indicating that net C uptake is highest in spring whereas C release is highest in summer (Laruelle et al., 2013). These estimates indicate that seasonal differences in shelf net C exchange are as high as the annually integrated latitudinal gradient. An analysis performed over Atlantic shelves suggests that the seasonal variability in the air-sea $CO_2$ exchange is most pronounced over temperate latitudes. In these regions, shelves generally

behave as strong $CO_2$ sinks in winter and spring, partly sustained by $CO_2$ fixation during the spring phytoplankton bloom, but can become mild $CO_2$ sources to the atmosphere in summer due to the effect of temperature-driven decrease $CO_2$ solubility in water (Laruelle et al., 2014). Such behaviour is consistent with that of the open ocean under similar latitudes (Takahashi et al., 2009).

Biogenic C emissions related to tropical aboveground biomass loss as well as crop and wood harvest are equally distributed

across months in this study. When exactly C emissions from humans and livestock occur is difficult to predict and would require more detailed consumption data (Wolf et al., 2015a).

### 4.4 Missing fluxes

Due to a focus on spatially explicit maps, not all known fluxes between land surface and atmosphere are considered in our analysis. We assume for instance that including the following fluxes may have an influence on the regional and global flux

estimates:



- Emissions from biogenic volatile organic compounds (VOCs) amount to approximately 0.76PgC / year (Sindelarova et al., 2014)

- $CO_2$ emissions from wetlands, estimated globally at around 2.1 PgC /year (Aufdenkampe et al., 2011)

- Crop residues burning in households

• Biofuel burning

- Changes in land management, e.g. shifts in agriculture, soil tillage, grassland ploughing and grazing

- Geological fluxes

- Raymond et al. (2013) estimate a much higher river evasion (1.8 PgC / year instead of 0.65 PgC / year used in this study).

**4.5    Uncertainty estimates and model-data fusion**

A comprehensive spatiotemporally explicit bottom-up estimate of NCE can be a powerful ingredient for model-data integration exercises (Rayner et al., 2005). Yet, model-data integration requires uncertainty characteristics of all used data streams (Raupach et al., 2005). Furthermore, it is important that uncertainties can be described in terms of random errors (Ciais et al., 2014). Error estimates at the local or regional level are difficult to use if no spatial error covariance matrix is

available. The uncertainty analysis presented in this study obtained through Monte Carlo sampling aims to be of use for modal-data-integration studies. Errors are automatically propagated through different spatial resolutions by aggregating the individual ensembles of NCE. Naturally, efforts should be made to obtain error estimates for all integrated datasets (i.e., Wood, Fires, Shelves, Estuaries, and Lakes). Nevertheless, this first integrated NCE estimate offers new possibilities for approaches such as the Carbon Cycle Data Assimilation System (CCDAS, Rayner et al., 2005), by not only providing a full

spatiotemporal grid of fluxes, but also a transparent and consistent error propagation scheme. This can have also practical applications, for instance for designing new measurement campaigns in regions with high uncertainties to reduce knowledge gaps in the global $CO_2$ fluxes.

**5    Outlook and conclusions**

If we exclude FF from the NCE estimate, we end up with a net $CO_2$ uptake by the Earth surface of 13.6±3.4 PgC / year. Assuming neutral $CO_2$ exchange for tropical forests (Sect. 3.2) still requires an additional source of about 4 PgC / year (2.5 PgC / year with river outgassing from Raymond et al., 2013), and potential candidates were suggested in Sect. 4.4. The estimate of 19 PgC / year for NEP seems rather high and in fact exceeds the estimate by Ciais et al. (in revision) over the RECCAP regions by nearly 9 Pg / year (7.5 Pg / year if the river outgassing by Raymond et al., 2013 is used). In particular,

our analysis probably strongly overestimates $CO_2$ uptake in South America, Africa Southeast Asia, and North America. A



comparison between upscaled ecosystem fluxes with a satellite-based inversion using GOSAT data lead to similar results (Kondo et al., 2015). One has to recall that the study aims for a transparent appraisal of the available and empirically derived pieces of information. It thus offers a quantitative approach to better identify knowledge gaps that should be addressed with future observation missions in to better constrain NCE in specific regions. In fact, we find that the uncertainty range of this

synthesis of data-driven $CO_2$ fluxes is still too large to provide any confidence for a budget based on bottom-up information only.

Relative uncertainties are comparably low in the most productive areas over land (tropical forests) in our synthesis, yet these are the largest contributors to the terrestrial net C flux and still largely unconstrained by ground-based observations (Tramontana et al., 2016). This overly high confidence indicates that the model ensemble spread does not capture the full

uncertainty and more appropriate methods are needed to quantify uncertainties in these products. As noted before, a sufficient network of ground-based observations of C exchange (i.e., eddy covariance towers) for constraining tropical C fluxes is still lacking (Baldocchi, 2014; Schimel et al., 2015).

Tough conditions for measurement campaigns in the Southern Ocean render it the largest undersampled ocean, leading to high uncertainties in the mean C exchange estimates (Landschützer et al., 2014; Rödenbeck et al., 2014). Plans for new

comprehensive sampling expeditions are underway (Newman et al., 2015).

As we have shown, the current observation-based bottom-up estimates are not yet precise enough to close the global C budget at the global scale. Yet at regional scale, high confidence can be achieved, for instance in Europe, Russia, East Asia, South Asia, and Australia. In addition, our estimates of NCE agree well with mostly independent estimates from other sources (Fig. 4). While in some region and for some subsystems more observational constraints are certainly necessary

(including the Southern Ocean, tropical land areas, and continental water bodies), new approaches to model respiratory and disturbance related fluxes are required to improve the upscaling of terrestrial ecosystem fluxes. Detailed information about soil properties, forest age and disturbance history is essential to improve the prediction of respiratory fluxes. Nevertheless, a purely data-derived bottom-up estimate of net C fluxes as presented here may be used as an additional constraint in model-data assimilation systems, to calibrate model parameters, or as priors for atmospheric inversions.

**Acknowledgements**

This study was funded by the European Union in the context of the FP7 project GEOCARBON (grant agreement #283080). Authors affiliated with [2] and [16] further acknowledge the EU for support via the H2020 project BACI (grant agreement #640176). Authors affiliated with [5] thank the SAMPLES project as part of the CGIAR research program CCAFS, and

CIFOR from the governments of Australia (grant agreement #46167) and Norway (grant agreement #QZA-10/0468). N.C. acknowledges funding from the NOVA grant UID/AMB/04085/2013. G.G.L. is 'Chargé de recherches F.R.S.-FNRS' at ULB. R.L. received funding from ANR (ANR-10-LABX-0018) and BRIC at ULB.



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



**Table 1. Datasets used in this study including reference, time period and number of ensemble runs. If not specified, temporal resolution is monthly.**

| Data set | Reference | Time period used | # Runs |
|---|---|---|---|
| Ocean | Landschützer et al. (2014) | 2001-2010 | 5+5 |
| | Rödenbeck et al. (2014) | | |
| Shelf | Laruelle et al. (2014) | 1 estimate | 1 |
| Estuaries | Laruelle et al. (2013) | 1 estimate | 1 |
| Marine | | 2001-2010 | 10 |
| Rivers | Lauerwald et al. (2015) | 1 estimate | 50 |
| Lakes | Raymond et al. (2013) | 1 estimate | 1 |
| GPP / TER | Tramontana et al. (2016) | 2001-2010 | 16 |
| Crops | Wolf et al. (2015b) | 2005-2010, annual | 10 |
| Wood | Poulter (2015) | 2000, 1 estimate | 1 |
| Fire | Giglio et al. (2013) | 2001-2010 | 1 |
| $E_{LUC}$ | Tyukavina et al. (2015) | 2000-2010, 1 estimate | 2 |
| | Harris et al. (2012) | 2000-205, 1 estimate | |
| FF (Fossil Fuels) | CARBONES | 2001-2010 | 1 |
| Atmospheric growth rate | NOAA | 2001-2010 | 1 |





**Table 2. Net carbon exchange for different subsystems and variables. Uncertainty is SD over ensemble runs, IAV is SD over annual values time (2001-2010), CV is coefficient of variation, computed as IAV over Mean.**

| Variable | Marine | Rivers | Lakes | -GPP | TER | Crops | Wood | $E_{LUC}$ | FF | Fire | NCE |
|---|---|---|---|---|---|---|---|---|---|---|---|
| Mean | -1.60 | 0.65 | 0.32 | -108.29 | 89.24 | 2.68 | 0.71 | 0.83 | 7.78 | 1.81 | -6.07 |
| Unc | 0.15 | 0.08 | | 3.62 | 3.60 | 0.21 | | 0.16 | | | 3.38 |
| IAV | 0.36 | | | 0.69 | 0. 33 | 0.09 | | | 0.75 | 0.11 | 0.63 |
| CV | 0.22 | | | 0.006 | 0.004 | 0.034 | | | 0.096 | 0.061 | 0.11 |





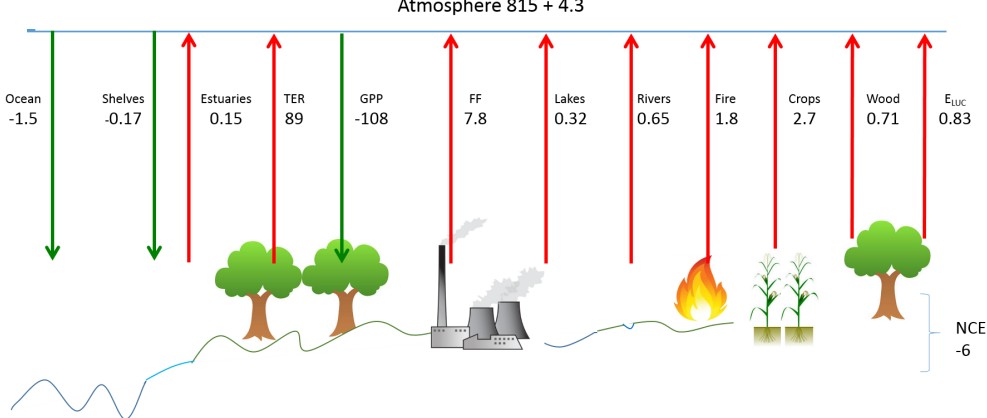

**Figure 1. Different components of observation-driven C exchange between the Earth's surface and the atmosphere. Red arrows denote a flux from the surface to the atmosphere (net source), green arrows denote a flux from the atmosphere to the surface (net sink). Units are in PgC / year.**





**Figure 2.**



Gridded spatial patterns of NCE. a) 2001-2010 decadal mean. b) Uncertainty; 1SD across the NCE ensemble. c) Relative uncertainty; uncertainty normalized by absolute mean. Latitudinal plots in b) and c) denote median across latitudes.

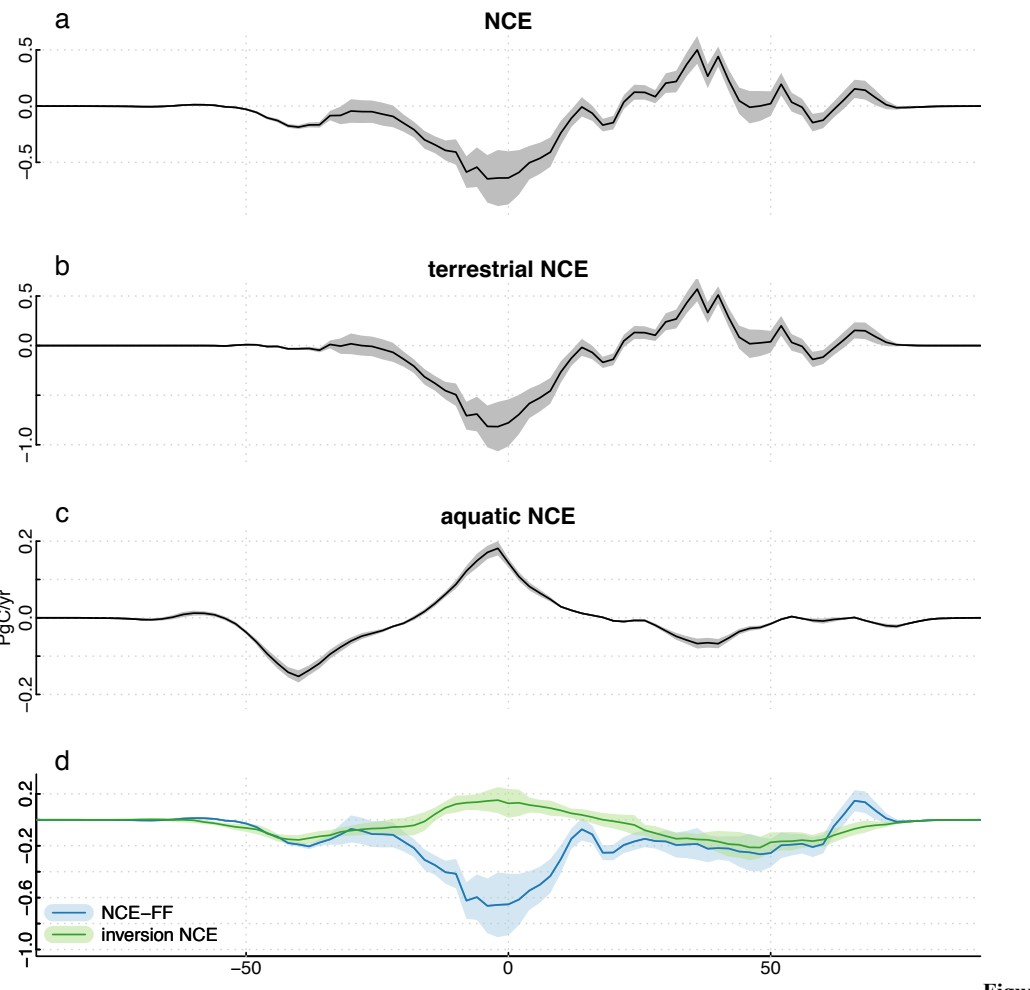

**Figure 3.** Mean and uncertainty (1 SD) of different subsets of NCE. a) All fluxes. b) Terrestrial fluxes. c) Aquatic fluxes. d) NCE without fossil fuels (blue) and NCE from inversions (by construction without fossil fuels, green).



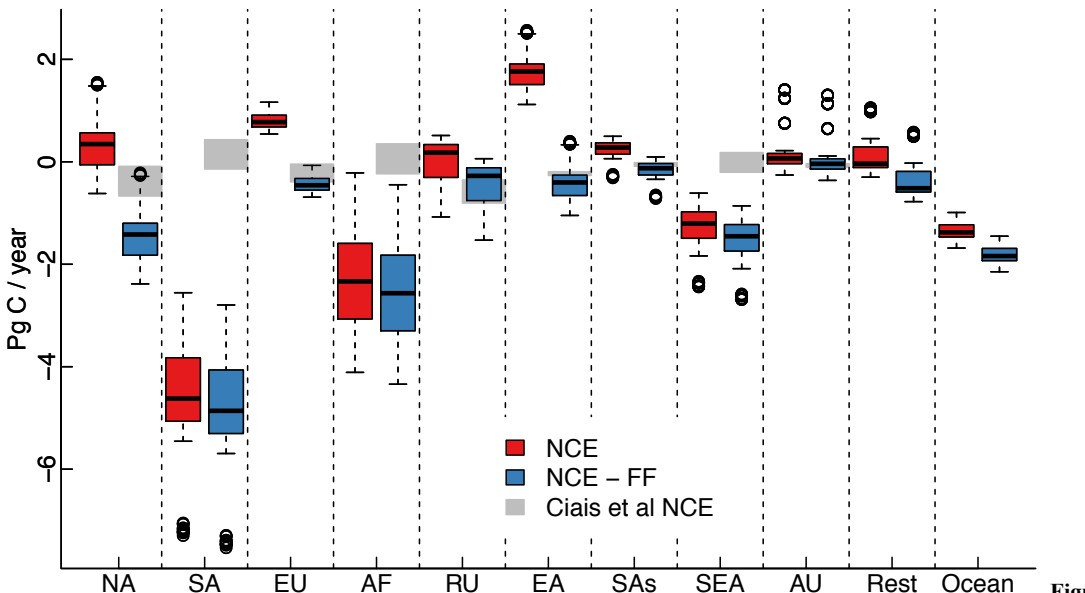

**Figure 4.**

**NCE over RECCAP regions, including (red) and without fossil fuels (blue). Shown are median, interquartile range (box) and 1.5 x interquartile range (whiskers). The regional estimates including uncertainties of NCE collected in Ciais et al (in revision) are underlain in grey.**





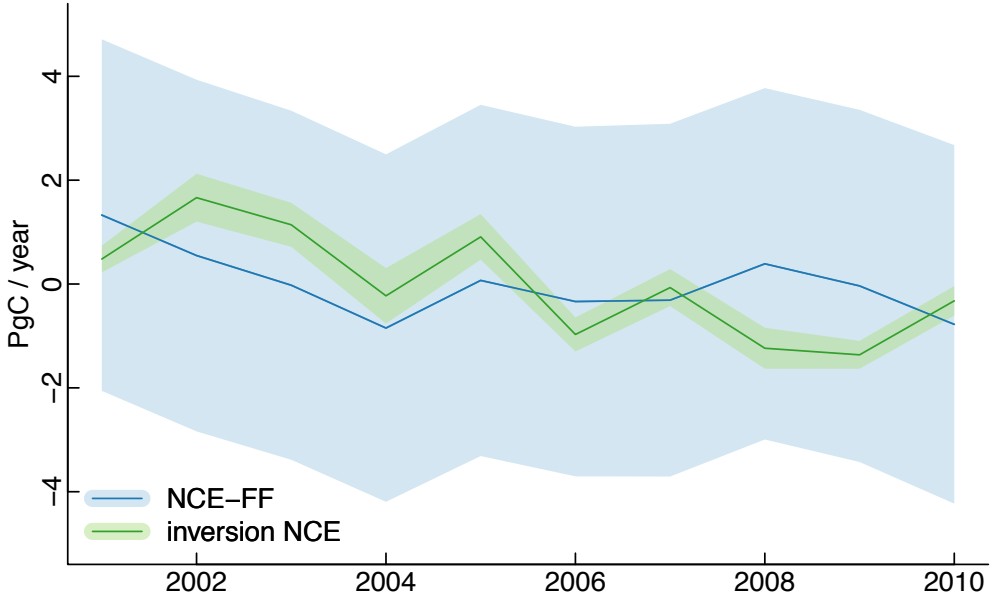

**Figure 5.**

**Comparison of NCE-FF with NCE from inversions (by construction without FF) on interannual time scales. Both time series were zero-centered by adding an offset of 13.87 PgC / year for NCE-FF and 3.74 PgC / year for NCE from inversions. Lines show mean, shading is 1 SD.**





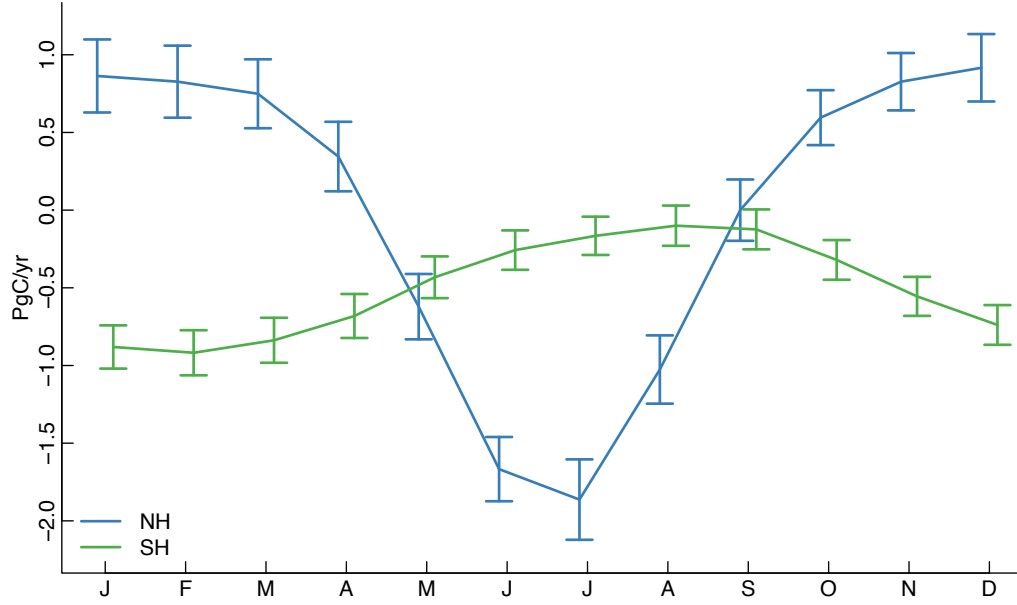

**Figure 6.**

**NCE mean seasonal cycle and uncertainty (1 SD) for Northern (blue) and Southern hemisphere (green).**