# Peer review of "An empirical spatiotemporal description of the global surfaceatmosphere carbon fluxes: opportunities and data limitations"

_Biogeosciences, 2016_

## Referee Comment (RC1) · Anonymous Referee #1 · 29 Nov 2016

Zscheischler et al. pull together a variety of surface to air $CO_2$ flux estimates and ask the question "Do these add up to a globally balanced budget?" This is a worthwhile effort, and the authors are using state of the art estimates. As alluded to in the text, the primary goal of this work is to create a combined data product that can be used as input to future data assimilation efforts.

Unfortunately, there are vital errors the analysis. Large annual cycles of $CO_2$ flux are taken into account for land, but entirely ignored for the ocean. The authors suggest that they are looking at the full "background" of natural $CO_2$ fluxes, but only consider the anthropogenic perturbation in the ocean. To be correct and consistent with the statements of a full accounting for natural background fluxes, Table 1 and Figure 1

should have large fluxes in the ocean that are of the order of the GPP and TER on land.

Furthermore, just assembling these data-based estimates into one with global coverage is not sufficient for publication. The analysis here is too thin, and the findings are poorly presented. Based on how the independent products have been produced, no one should expect that they would add up to a balanced budget – this finding is no surprise. The authors do not do enough to explain what are the major sources of the uncertainty, nor do they do enough to make it clear how they estimate this uncertainty. They need to do a lot more with the products that they have before this manuscript is acceptable for publication.

Major Comments

1. The authors indicate that their goal is to not just address anthropogenic carbon uptake, but to also address the background carbon fluxes (Page 3). Yet their methodology is inconsistent across land vs ocean in this respect. While on land, they separate GPP uptake of $CO_2$ from TER efflux, they completely ignore the comparable cycle in the ocean. See Figure 6.1 of Ciais et al. 2013 (IPCC WG1, Chapter 6) where it is clear that the naturally occurring cycle in the ocean creates an exchange flux of 80 PgC/yr out of the ocean and 78.4 PgC/yr into the ocean; this is comparable in magnitude to the GPP and TER (+- ~100 PgC/yr), but the authors here simply ignore these ocean fluxes by only presenting their sum. They also appear to ignore these large fluxes in their assessment of uncertainty (though detail on how uncertainty is accounted for is so thin that it is hard for the reviewer to be sure on this point). The full background cycle in the ocean must be included in this analysis must be remedied in this analysis.

2. A coherent explanation for the large imbalance in the final "budget" is never presented, instead the reader is left is a laundry list (e.g. page 16) of possibilities and no clarity of what the authors have identified as the likely most important uncertainties. It seems quite likely that the large GPP and TER fluxes, or the comparable ocean fluxes,

are biased high or low. Their uncertainties are the only ones on the same order as the NCE uncertainty. This issue is even more obvious at the regional scale. This issue should be more directly addressed.

a. One clear place to do this would be on Page 12, where it is stated that in 13% of their runs, the global C source is consistent with the atmospheric growth rate. If this finding is meaningful, these 13% of runs need to be analyzed and presented clearly so that the reader can understand what is different about them. A simple explanation for the uncertainty in the budget could be that GPP is overestimated by 10%, and if all of these 13% of runs have GPP on the low side, then it would be useful to identify such a pattern.

3. There are many inconsistencies in the data products used here. For example, for the ocean flux the parameterization of gas exchange is Wanninkhof (1992) with ERA-interim winds, but for the shelf it is Wanninkhof et al. (2013) with CCMP winds. These differences could make a significant difference to the ultimate fluxes even though based on the same pCO2 database. On the one hand, this reviewer recognizes that these differences are due to choices made by the providers of these previously-published flux products, and cannot be easily changed by these authors. Nevertheless, some evaluation of these effects should be performed. One possibility for such evaluation could be in the overlap regions of the three products that go into the merged Marine flux field.

4. The text is difficult to follow, particularly in the discussion and conclusion sections. These sections read as a list of possibilities, without clarity as to what is really important. The authors need to do more to provide this needed clarity.

Minor comments

- Line 15 "limitations."

[Figure]

- Line 24 Which regions have the large net sink? The authors specify several regions with flux to the atmosphere, then the sum of all is large and negative, presumably due to the tropics. The reader should be able to better understand where this large negative is coming from geographically based on the abstract.

- Line 15: "background CO2 fluxes over land and ocean," This analysis only accounts for background fluxes in the land, not in the ocean. Instead this analysis suggests there is no net background ocean flux, only the anthropogenic residual! In contrast to Figure 6.1 of IPCC WG1 (Ciais et al. 2013), this analysis ignores background, natural ocean exchanges. This is a major error that must be remedied.

- Line 2 What is the meaning of "resampled". Is this averaging of all points in a 1x1 grid? If data are at coarser resolution, what is done? Please be more specific. Show that global mean values of the variables considered are conserved by this method.

- Line 25 "For NCE estimates, we randomly combined all datasets, using a single realization of each flux, to generate an estimate of NCE." What is the meaning of "randomly combined all datasets"? How is this random if all datasets are used? If the "random combination" applies only to the 2 fluxes (ocean and LUC) that have multiple sources according to Table 1, then the result here is an incomplete estimate of uncertainty. More explanation is needed here so that the reader can have confidence in the uncertainty estimate being made.

- Overall it is hard to understand how the uncertainty is propagated. Bits and pieces are mentioned under each of the flux products below, but a coherent picture is not made clear. Perhaps this lack of clarity could be partially remedied with a schematic figure that clarifies how many different realizations of each flux and how the sampling across them is performed.

- Line 16 "schused" is a typo

- Line 15 If the same FLUXCOM product is being used to separate the gpp and ter, the ocean fluxes out vs in should also be separated (Figure 6.1, Ciais et al. 2013). It is inconsistent to take different approaches with land vs ocean, and skews the reader impression of the magnitude of local fluxes and their uncertainty.

- Line 8: A reference for EDGAR is likely warranted.

- Line 24 "Not all inversions were available till 2010." What is done if this is the case?

- Line 6: That this imbalance is not real, but an artefact of the uncertainty of the data should be made explicitly clear here; not just left for section 4.

- Line 12: "whereas in fact many errors might be correlated as this is clearly the case for GPP and TER." The same statement will almost certainly be appropriate in the case of uncertainty in ocean fluxes, once they are appropriately accounted for.

- Line 22: "Due to the small contribution of the oceans, absolute uncertainties are barely discernible." Comment: This will probably be different once out vs in is considered separately.

- Line 14 "We use the land cover map of 2005 from the European Space Agency (http://www.esa-landcover-cci.org/) to identify tropical forests (all pixels where broadleaved evergreen trees dominate). " Why use satellite product, when FLUXCOM model is what your estimate is based on. The FLUXCOM land cover product should

be used.

- Line 3-11: This section overstates the level of agreement with Ciais et al. (in revision). It suggests that Figure 3 illustrates "good agreement" except for a few regions, but when one looks closely, only 5 land regions agree, including Australia that is basically zero, while 4 do not. Overstatement is exemplified by this sentence "Given that Ciais et al. (revision) rely on an independent method, this demonstrates that a good understanding of net C fluxes exists for non-tropical areas, North America excluded." This section should be written more carefully to acknowledge that lack of agreement is as common as agreement.

- Line 6: NEP should be defined again as its not been defined for many pages.

- Line 23-end: This reads as it may be one hypothesis of many. Or is it a leading one? The reader needs the authors to be more clear.

- Line 8: "often not too far off but given that different top-down studies using different atmospheric models provide conflicting information on the adjustments needed to align modelled concentrations with measured ones, this information cannot be used to provide clear uncertainty ranges." This is not understandable to someone doesn't work with these models or know this jargon.

- Line 13 "To better constrain C exchange on a monthly basis, however, the seasonal cycles of those fluxes would be necessary." Comment: Awkward phrasing.

- Line 23 "at similar latitudes."

- The degree to which the unaccounted fluxes (list) could be large enough to impact the global "budget" should be discussed. Each of these fluxes should be quantified to the best degree possible so as to put in context with the overall budget. The laundry list approach is not helpful to the reader, particularly when so many of the proposed fluxes are left entirely unquantified, and the authors do not discuss the list at all after it is presented. What is the reader to meant to conclude?

- Line 14: Regions such as the North Atlantic (Schuster et al. 2013, Biogeosciences) should also be noted as having large uncertainty at seasonal timescales and beyond.

Section 5 overall:

- This section is also poorly organized. It reads as a listing of issues largely already mentioned prior. It needs to be rewritten to focus on the key findings of this work – What are the take-home messages that the reader should be getting?

Table 2: - should note that negative is from the atmosphere.

- If the label is –GPP then GPP should be 108.29 not -108.29

- A consistent number of significant figures should be used, unless the authors can justify the greater precision of the numbers with 5 significant figures (GPP) as opposed to those with only 2 or 3. This is important because it is uncertainty in GPP that drives most of the NCE uncertainty. The GPP numbers is clearly not actually known to 5 significant figures.

- All numbers should have the same fontsize, or if the different sizes have a meaning, it should be noted

- The full decomposition of the "marine" should be noted in this table, so as to be consistent with Figure 1

- The natural fluxes of the ocean need to be accounted for in a manner comparable to

GPP and TER.

Figure 1

- The units on the 815 are presumably PgC. This should be noted explicitly on the figure or in the caption.

- The ocean should have two arrows, one in and one out. The picture from this figure should be consistent with Figure 6.1 of IPCC in that both the ocean and the land have a large background, natural cycle on top of which the anthropogenic is superimposed.

Figure 2

- The colorbar in panel a is mislabeled as "%"

Figure 3

- The x-axis needs a label

Figure 4

- What are the circles? Presumably outliers? Clarify in caption.

- The regions indicated by each acronym should be noted in the caption.

---

## Referee Comment (RC2) · Anonymous Referee #2 · 23 Dec 2016

This paper puts together a wide range of spatially explicit bottom-up surface-atmosphere CO2 flux data sets aiming to reconcile the carbon budget from bottom-up estimation and the atmospheric CO2 growth rate. While this type of research is needed for improving our understanding of carbon cycle, this study has serious flaws in generating the data and is lack of validation and deep analysis of the combined data set. The language is vague in many places. At this stage, I don't recommend publishing the paper.

Major comments: 1. The added value of the new combined dataset is very limited. The authors simply put different data streams together, and there is no effort trying to harmonize the data, even though some of the datasets do not cover the same time

period, e.g., the crops cover 2005 to 2010.

2. The paper compares the bottom-up estimations with the top-down inversion results (Figures 3 and 4, section 3.4), but it is lack of discussion about why these two approaches have different results, and which estimation is closer to reality.

3. In section 3.4, it says that "both estimates agree well in the extratropics", but the figure 3d shows that the NCE-FF and atmospheric inversion results also have large differences in the NH high latitudes (between ~60N and ~75N), with the NCE-FF indicating a source to the atmosphere, while the atmospheric inversion indicating a weak sink.

4. Even though the latitudinal pattern of the inversion results follows a pattern similar to that of the aquatic fluxes (Figure 3c), there is no direct evidence indicating the propagation of the marine signal into continents during atmospheric inversion. I suggest removing the discussion on the pattern comparison between aquatic fluxes and atmospheric flux inversion results in section 3.4.

5. Section 4.5 discusses the possible application of the combined dataset in model-data integration studies. It is an interesting idea. However, with such large uncertainties (with more than 10GtC disagreement with the atmospheric CO2 growth rate) in the combined dataset and a mixture of all different carbon flux components, it is not clear how such product can be used in carbon cycle data assimilation that focuses primarily on land carbon fluxes. What is the added value of using such data set compared to directly using flux tower observations? In addition, if such product were to be used as "observations" in a data assimilation system, a rigorous validation against independent observations is needed.

6. In section 3.2, it says that "13% of our runs we obtain a global C source that is consistent with the atmospheric growth rate", what are the spatial distributions of the fluxes from these 13% runs?

7. L26 (P14): what is the distribution of the different age classes of forests in FLUXNET? Is there solid evidence showing that the year and regrowing forests are overrepresented in FLUXNET?

8. Section 3.5 discusses seasonal cycle and monthly variability. It would be helpful to put this discussion in perspective, e.g., comparing to other independent estimations, so that the readers would know the credibility of this result. It is not clear what are the latitude ranges for the NH and SH in Figure 6.

9. Line 6 (p15), what is the basis for the 50% uncertainty?

Minor comments

1. In the abstract, "would require an offsetting surface C source of 4.27 $\pm$ 0.10 PgC/yr", should the offset be 4.27 + 6.07 PgC/yr in order to have 4.27 PgC atmospheric CO2 growth rate?

2. Line 13-16 (p3), it is not clear what the "background CO2 fluxes" means.

3. Line 23 (p4), "goal of this study the" should be "goal of this study to"

4. Line 16 (p6): what is "schused"?

5. Line 21 (p14): What does the "relevant drivers" refer to? Be more specific.

6. Line 29 (p14): what does the "global driver" refer to?

---

## Author Comment (AC1) · 27 Jan 2017

*We would like to thank reviewer #1 for the detailed review of our manuscript and the thoughtful suggestions that will help to improve our manuscript. In the following, we will answer each of the reviewers comment.*

Zscheischler et al. pull together a variety of surface to air $CO_2$ flux estimates and ask the question "Do these add up to a globally balanced budget?" This is a worthwhile effort, and the authors are using state of the art estimates. As alluded to in the text, the primary goal of this work is to create a combined data product that can be used as input to future data assimilation efforts. Unfortunately, there are vital errors the analysis. Large annual cycles of $CO_2$ flux are taken into account for land, but entirely ignored for the ocean.

*We believe that that this is a misunderstanding. We do consider annual cycles of all variables – where these were available. And these were indeed available for most land and ocean area. State-of-the-art observation-based estimates of shelf areas and inland waters are however still missing the seasonal representation. This kind of gap analysis is exactly what we intend to do here: demonstrate where we currently miss information to achieve a comprehensive and purely data driven description of the surface-atmosphere $CO_2$ exchange. We believe that this is the best way forward to improve our future understanding and fill these knowledge gaps.*

The authors suggest that they are looking at the full "background" of natural $CO_2$ fluxes, but only consider the anthropogenic perturbation in the ocean. To be correct and consistent with the statements of a full accounting for natural background fluxes, Table 1 and Figure 1 should have large fluxes in the ocean that are of the order of the GPP and TER on land. Furthermore, just assembling these data-based estimates into one with global coverage is not sufficient for publication.

*To the best of our knowledge this is the first critical appraisal of data driven estimates of surface-atmosphere $CO_2$ fluxes, which may be relevant for wide community working in C cycle science. For the ocean we provide estimates of the contemporary carbon fluxes, i.e. a combination of natural and anthropogenic fluxes. Based on surface ocean $pCO_2$ observations the natural and anthropogenic components cannot be separated (see also below).*

The analysis here is too thin, and the findings are poorly presented. Based on how the independent products have been produced, no one should expect that they would add up to a balanced budget – this finding is no surprise.

*Indeed, we don't expect the readers to be surprised that individual components do not add up – but we assume that e.g. the spatially explicit description of the data uncertainty and budget mismatch is of key importance to guide future research efforts. I.e. while it is no surprise that the numbers don't add up, it is of much more importance where they don't add up and where we have today the largest observational knowledge gaps and uncertainties.*

The authors do not do enough to explain what are the major sources of the uncertainty, nor do they do enough to make it clear how they estimate this uncertainty.

*We apologize if we didn't achieve a sufficient description of how we estimate uncertainty. To improve the presentation and to make it more clear how we derive our uncertainties, we will adjust the text and include a visual description of the work flow (see illustration at the end of this document).*

They need to do a lot more with the products that they have before this manuscript is acceptable for

publication.

*We agree, that the data we present here would allow much more analysis and we would encourage the community to use the datasets and add additional analysis. Overall, however, we would like to reemphasize that the main aim of this study is not to simply check whether data-driven surface-atmosphere $CO_2$ fluxes add up to a balanced budget. Given the difficulties of guaranteeing a consistent and contiguous global C monitoring system, this simply cannot be expected. And the current data-driven knowledge about many of the relevant fluxes cannot compensate this. But – and we find this an important contribution – we provide global spatiotemporal estimates of the net carbon flux combining a variety of heterogeneous datasets and consistently propagate their uncertainties. Through our approach we can identify regions of high and low uncertainty, guiding new monitoring campaigns and novel scientific approaches to reduce specific uncertainties. Our NCE estimates specify contemporary fluxes over the whole Earth surface, thus including background and anthropogenic fluxes, as explained below.*

Major Comments
1. The authors indicate that their goal is to not just address anthropogenic carbon uptake, but to also address the background carbon fluxes (Page 3). Yet their methodology is inconsistent across land vs ocean in this respect. While on land, they separate GPP uptake of CO2 from TER efflux, they completely ignore the comparable cycle in the ocean. See Figure 6.1 of Ciais et al. 2013 (IPCC WG1, Chapter 6) where it is clear that the naturally occurring cycle in the ocean creates an exchange flux of 80 PgC/yr out of the ocean and 78.4 PgC/yr into the ocean; this is comparable in magnitude to the GPP and TER (+-100 PgC/yr), but the authors here simply ignore these ocean fluxes by only presenting their sum. They also appear to ignore these large fluxes in their assessment of uncertainty (though detail on how uncertainty is accounted for is so thin that it is hard for the reviewer to be sure on this point). The full background cycle in the ocean must be included in this analysis must be remedied in this analysis.

*The reviewer is correct that we have only displayed aquatic net fluxes (not only for the open ocean but throughout the whole aquatic system), which was simply an effect of data availability. We also concur that the gross fluxes may have individually larger uncertainties attached to them. We do, however, disagree, that we „ignore" these fluxes or their uncertainty. The uncertainty of the net flux presented in this study is comparable to other uncertainty estimates such as the IPCC report or the Global Carbon Budget (e.g. Le Quéré et al 2015). The reason for the smaller net uncertainty stems from the correlation between air-sea and sea-air fluxes and their uncertainty. The largest source of uncertainty between individual flux elements (sea-air or air-sea) stem from the gas transfer formulation (see also answer to a more specific comment below), i.e., a systematic source of uncertainty which likewise effects fluxes in both direction leading to a much smaller net flux difference and attached uncertainty. This is also stated in the caption of the IPCC mentioned by the reviewer: "Individual gross [air–sea exchange] fluxes and their changes since the beginning of the Industrial Era have typical uncertainties of more than 20%, while their differences (Net land flux and Net ocean flux in the figure) are determined from independent measurements with a much higher accuracy (see Section 6.3). Therefore, to achieve an overall balance, the values of the more uncertain gross fluxes have been adjusted so that their difference matches the Net land flux and Net ocean flux estimates."*

*Over land we have used TER and GPP because these fluxes are available at the spatiotemporal grid which we required. This is not the case for the ocean. However, we agree with the reviewer that this is inconsistent. In the revised version, we will therefore only use the directly upscaled NEP product from FLUXCOM, which will reduce the sample size of the NEP estimates from 16 to 8 (the uncertainty related to the flux separation (split of NEP into GPP and TER) will be dropped, as it is not relevant for*

*the uncertainty estimation of the net fluxes). The uncertainty in the upscaled NEP product is 2.1PgC/yr (compared to 3.4PgC/yr when using TER-GPP), which is still much larger than the uncertainty of the net flux over the ocean (0.15PgC/yr). Using directly upscaled NEP leads to similar global mean estimates than using TER-GPP (the difference is <0.7PgC/yr, i.e., <5%).*

*In response to the reviewer's comments we will also add above explanation regarding net uncertainty in the text. We do however avoid the term „background fluxes" as this term can be easily confused with „natural fluxes", whereas we cannot separate natural and anthropogenic components. All our flux estimates are the aggregates of natural fluxes and anthropogenic disturbance. We address this in more detail below, in direct response to an additional comment by this referee.*

2. A coherent explanation for the large imbalance in the final "budget" is never presented, instead the reader is left is a laundry list (e.g. page 16) of possibilities and no clarity of what the authors have identified as the likely most important uncertainties. It seems quite likely that the large GPP and TER fluxes, or the comparable ocean fluxes, are biased high or low. Their uncertainties are the only ones on the same order as the NCE uncertainty. This issue is even more obvious at the regional scale. This issue should be more directly addressed.

*In the revised version we will more clearly emphasize and discuss the most likely reasons for this imbalance. In our opinion it is most likely a combination of*
  *i)       a bias in NEP, most probably in the tropics where only very few eddy covariance sites lead to a weak observational constraint.*
  *ii)      missing sources (as listed in section 4.4), especially emissions from wetlands and VOCs.*

a. One clear place to do this would be on Page 12, where it is stated that in 13% of their runs, the global C source is consistent with the atmospheric growth rate. If this finding is meaningful, these 13% of runs need to be analyzed and presented clearly so that the reader can understand what is different about them. A simple explanation for the uncertainty in the budget could be that GPP is overestimated by 10%, and if all of these 13% of runs have GPP on the low side, then it would be useful to identify such a pattern.

*We have thought about this suggestion but concluded that this would imply a level of insight that is not supported by the data, i,e, the large uncertainty related to missing fluxes (see page 16 and comment above). Alternatively, constraining the NEP ensemble could thus be misleading because the constraint is highly uncertain. Furthermore, if we assume that relevant drivers are missing in the set of predictors in FLUXCOM (e.g. forest age, see Sect. 4.1), all members in FLUXCOM are biased in the same way (all members use the same set of predictors). Constraining the ensemble, even if we had a well-defined constraint, thus cannot provide us with new insights into the processes that need further investigation or drivers that are missing in the set of predictors. For these reasons we decided to omit the part where we state that 13% of the NCE runs are consistent with the CGR.*

3. There are many inconsistencies in the data products used here. For example, for the ocean flux the parameterization of gas exchange is Wanninkhof (1992) with ERA-interim winds, but for the shelf it is Wanninkhof et al. (2013) with CCMP winds. These differences could make a significant difference to the ultimate fluxes even though based on the same pCO2 database. On the one hand, this reviewer recognizes that these differences are due to choices made by the providers of these previously-published flux products, and cannot be easily changed by these authors. Nevertheless, some evaluation of these effects should be performed. One possibility for such evaluation could be in the overlap regions of the three products that go into the merged Marine flux field.

*We agree that there are inconsistencies between the data products. As noted by the referee, it is not the aim of the study to re-calculate estimates, but rather bring together existing knowledge. We have tried to account for many inconsistencies, i.e., we calculated flux estimates at the same spatial and temporal resolution, we have unified the uncertainty calculation procedure, we have recalculated overlap areas to avoid double accounting, etc, but as rightfully noted by the referee, there are still some other sources we have not accounted for. The referee highlights the gas exchange formulation as an example and we concur that this is a factor that has a significant impact on the air-sea exchange of CO2. However, as explained above, the net effect of the gas flux formulation is of lesser importance when the net flux is considered. We would also like to note that while the open ocean estimates use the formulation of Wanninkhof et al 1992, i.e., a quadratic dependency between gas flux and wind speed at 10m, they use more recent gas transfer coefficients (Rödenbeck et al and Landschützer et al scale their estimates to match a mean transfer velocity of 16cm/hr as suggested by Wanninkhof et al 2013). However, we concur with the reviewer that there is an additional uncertainty related to the transfer. In this way, uncertainty in ocean estimates is probably underestimated. This is, however, also true for land based estimates. E.g. in FLUXOM, all models use the same set as predictors. We welcome the suggestion of the reviewer to use the overlap area for testing, however, this overlap area is very local and is biased towards coastal zones. We will discuss this probable underestimation of uncertainty in the discussion section. Landschützer et al 2014 estimated that the choice of transfer formulation and the pCO2 mapping mismatch (also including other relationships than quadratic) lead to an uncertainty of 37% for the global average over 1998-2011, with the majority of this uncertainty stemming from the gas transfer formulation.*

4. The text is difficult to follow, particularly in the discussion and conclusion sections. These sections read as a list of possibilities, without clarity as to what is really important. The authors need to do more to provide this needed clarity.

*We will reformulate these sections to emphasize and discuss our main results better. In particular, we will focus the conclusion on these 3 main results:*
  i)     *Current spatiotemporally explicit observation-driven estimates of surface-atmosphere CO2 exchange are not constrained well enough to close the carbon budget at the global scale.*
  ii)    *Regionally, those estimates are partly well constrained and may be used for model-data integration studies and validation of models. These regions include Europe, Russia, South Asia, East Asia, Australia and many oceanic regions. Constraining C fluxes in regions with currently high uncertainties better should be a priority of future research.*
  iii)   *The most likely candidate for inducing the mismatch between data-driven surface-atmosphere CO2 exchange and the atmospheric CO2 growth rate is land NEP, in particular tropical NEP estimates appear to be very uncertain. Understanding this bias will help designing better upscaling approaches (e.g. by including currently missing relevant drivers) and pinpointing variables that need to be (better) monitored in the future.*

Minor comments
- Line 15 "limitations."

*Thanks.*

- Line 24 Which regions have the large net sink? The authors specify several regions with flux to the atmosphere, then the sum of all is large and negative, presumably due to the tropics. The reader should

be able to better understand where this large negative is coming from geographically based on the abstract.

*The large C is over found over most of the tropical land, i.e., Amazon, Congo and Indonesia. We will mention these regions in the revised abstract.*

- Line 15: "background CO2 fluxes over land and ocean," This analysis only accounts for background fluxes in the land, not in the ocean. Instead this analysis suggests there is no net background ocean flux, only the anthropogenic residual! In contrast to Figure 6.1 of IPCC WG1 (Ciais et al. 2013), this analysis ignores background, natural ocean exchanges. This is a major error that must be remedied.

*We believe that this is a misunderstanding. The provided flux estimate is neither natural nor anthropogenic but the contemporary flux, i.e., a combination of both natural and anthropogenic. Using surface ocean observations to estimate our flux we cannot distinguish between natural and anthropogenic. To separate the components, we would need to know the pre-industrial state of the ocean. Some budgets, e.g. the global carbon budget solve this e.g. by using an estimate of pre-industrial net ocean outgassing of 0.45PgC/yr (Jacobsen et al 2007) that is derived from riverine carbon input and prohibits CO2 saturation between the ocean and the atmosphere. However, this number does not account for natural variability of the pre-industrial flux. The study also does not focus on the natural carbon budget, or on the anthropogenic flux budget, but rather tries to quantify contemporary air-sea and air-land fluxes based on available observations. The same holds for the land. To make this clearer we will clarify this aspect in the introduction of the revised manuscript.*

- Line 2 What is the meaning of "resampled". Is this averaging of all points in a 1x1 grid? If data are at coarser resolution, what is done? Please be more specific. Show that global mean values of the variables considered are conserved by this method.

*All datasets have at least 1x1 degree resolution (1x1 degree or finer), such that resampling here means averaging to this coarser resolution. All global means are conserved by this averaging (by taking the land-ocean masks into account). We will explain this in more detail in the methods section of the revised manuscript.*

- Line 25 "For NCE estimates, we randomly combined all datasets, using a single realization of each flux, to generate an estimate of NCE." What is the meaning of "randomly combined all datasets"? How is this random if all datasets are used? If the "random combination" applies only to the 2 fluxes (ocean and LUC) that have multiple sources according to Table 1, then the result here is an incomplete estimate of uncertainty. More explanation is needed here so that the reader can have confidence in the uncertainty estimate being made.

*The random combination applies always to all fluxes contributing to NCE. That is, we create multiple estimates of NCE by summing up different random combinations of the source datasets following Eq. 1. Uncertainty is then estimated based on the newly generated NCE ensemble. See also the response to the comment below.*

- Overall it is hard to understand how the uncertainty is propagated. Bits and pieces are mentioned under each of the flux products below, but a coherent picture is not made clear. Perhaps this lack of clarity could be partially remedied with a schematic figure that clarifies how many different

realizations of each flux and how the sampling across them is performed.

*We will introduce a schematic figure to better explain how the uncertainties are propagated, as the reviewer suggested (see Figure R1 below). Each of the 200 NCE ensemble members consists of the sum of randomly selected members of the fluxes contributing to NCE (see Eq. 1). In principle, we could create 10\*10\*50\*16\*10\*2=160000 different NCE estimates by combining all the available members (see the #Runs shown in Table 1). We limit ourselves to 200 NCE estimates as a representative sample for the whole distribution due to the prohibitive computational expense of running all 160000 combinations. All NCE uncertainty estimates are then derived from these 200 runs. This approach implicitly contains information on the spatiotemporal uncertainty structure of the NCE estimates (i.e., the error covariance matrix). In this way regional or continental NCE uncertainties can be computed by aggregating each of the 200 NCE estimates over the desired region, automatically taking correlated errors into account. This was for example done for Figure 4.*

- Line 16 "schused" is a typo

*Thanks, should be "used" and will be changed in the revised version.*

- Line 15 If the same FLUXCOM product is being used to separate the gpp and ter, the ocean fluxes out vs in should also be separated (Figure 6.1, Ciais et al. 2013). It is inconsistent to take different approaches with land vs ocean, and skews the reader impression of the magnitude of local fluxes and their uncertainty.

*As explained above, in the revised version we will only use the directly upscaled NEP product over land to be consistent and to focus the attention on the uncertainties that are related to the net fluxes only.*

- Line 8: A reference for EDGAR is likely warranted.

*Thank you, we will insert the reference in the revised version.*

- Line 24 "Not all inversions were available till 2010." What is done if this is the case?

*The mean and uncertainty for each year is taken over all available inversions for that year. We will specify this in the revised version.*

- Line 6: That this imbalance is not real, but an artefact of the uncertainty of the data should be made explicitly clear here; not just left for section 4.

*We agree with the reviewer that this statement may be misunderstood. We report the mismatch which is obtained when combining all currently available spatiotemporal data-driven surface-atmosphere fluxes. This statement thus highlights that our current knowledge on C fluxes is not sufficient to close the C budget in this way (i.e., leaving out process-based models). We will add a comment on this in the revised version.*

- Line 12: "whereas in fact many errors might be correlated as this is clearly the case for GPP and TER." The same statement will almost certainly be appropriate in the case of uncertainty in ocean fluxes, once they are appropriately accounted for.

*As stated above, the uncertainties in the gross fluxes over the ocean are highly correlated, leading to much smaller uncertainties for the net fluxes (see also the caption of the IPCC figure mentioned by the reviewer). By using the ocean net flux estimates for NCE, these correlated uncertainties are automatically accounted for. As stated before, in the revised version we will only use directly upscaled NEP over land. Hence the comparison of the uncertainties will be more consistent.*

- Line 22: "Due to the small contribution of the oceans, absolute uncertainties are barely discernible." Comment: This will probably be different once out vs in is considered separately.

*As outlined in our response to the reviewer's major comment above, the net sink uncertainty is in fact much smaller than air-sea and sea-air flux uncertainty, due to the correlation between the uncertainties between the individual components. Using NEP over land (instead of TER-GPP) will decrease this difference slightly.*

- Line 14 "We use the land cover map of 2005 from the European Space Agency (http://www.esa-landcover-cci.org/) to identify tropical forests (all pixels where broadleaved evergreen trees dominate). " Why use satellite product, when FLUXCOM model is what your estimate is based on. The FLUXCOM land cover product should be used.

*Thank you for this comment, we will use the map of plant functional types used in FLUXCOM in the revised version.*

- Line 3-11: This section overstates the level of agreement with Ciais et al. (in revision). It suggests that Figure 3 illustrates "good agreement" except for a few regions, but when one looks closely, only 5 land regions agree, including Australia that is basically zero, while 4 do not. Overstatement is exemplified by this sentence "Given that Ciais et al. (revision) rely on an independent method, this demonstrates that a good understanding of net C fluxes exists for non-tropical areas, North America excluded." This section should be written more carefully to acknowledge that lack of agreement is as common as agreement.

*We agree with the reviewer and will formulate this more carefully in the revised version, better highlighting the large uncertainties.*

- Line 6: NEP should be defined again as its not been defined for many pages.

*Thanks, will be done in the revised version.*

- Line 23-end: This reads as it may be one hypothesis of many. Or is it a leading one? The reader needs the authors to be more clear.

*This is the leading one. We will emphasize this more in the revised version. To test this hypothesis rigorously, the complete upscaling procedure needs to be redone with including a forest age map as a*

*predictor, which should be done in future research.*

- Line 8: "often not too far off but given that different top-down studies using different atmospheric models provide conflicting information on the adjustments needed to align modelled concentrations with measured ones, this information cannot be used to provide clear uncertainty ranges." This is not understandable to someone doesn't work with these models or know this jargon.

*We will rephrase the section regarding uncertainties in fire emission referring to a recent paper on global fire emissions by van der Werf et al, currently in discussion for Earth System Science Data (van der Werf et al., 2017). Here it is stated that assuming 50% uncertainty overall is a best guess assessment, and better quantifying this uncertainty requires an assessment of the burned area estimates as well as new field data on fuel consumption and emission factors. We cannot, however, propagate this uncertainty to the NCE estimates as this would require spatiotemporal error covariance matrices.*

- Line 13 "To better constrain C exchange on a monthly basis, however, the seasonal cycles of those fluxes would be necessary." Comment: Awkward phrasing.

*We will rephrase this statement as "Estimates of the seasonal variation in these fluxes are necessary to better constrain seasonal variations in NCE."*

- Line 23 "at similar latitudes."

*Thanks.*

- The degree to which the unaccounted fluxes (list) could be large enough to impact the global "budget" should be discussed. Each of these fluxes should be quantified to the best degree possible so as to put in context with the overall budget. The laundry list approach is not helpful to the reader, particularly when so many of the proposed fluxes are left entirely unquantified, and the authors do not discuss the list at all after it is presented. What is the reader to meant to conclude?

*Thank you for this comment. We have provided estimates for those fluxes which have been quantified in the past. The remaining fluxes can be assumed to be rather minor, though little is currently known. We will discuss how much of the obtained global mismatch could be remedied by those fluxes in the revised version. Due to this additional uncertainty, we cannot exclude FLUXCOM runs with particularly high carbon uptake.*

- Line 14: Regions such as the North Atlantic (Schuster et al. 2013, Biogeosciences) should also be noted as having large uncertainty at seasonal timescales and beyond.

*The RECCAP initiative has shown that the largest uncertainties are in the southern hemisphere, i.e., in Schuster et al 2013, the South Atlantic has shown much less agreement between methods than the North Atlantic when low frequency signals – such as IAV and trends – are considered, whereas, methods generally agree seasonally where the seasonal cycle is dominated by the temperature variability, i.e., subtropics. In general, the ocean RECCAP studies (Sarma et al, Ishii et al, Schuster et al, Lenton et al and Wanninkhof et al) have shown regionally substantial differences between, however, few of these papers provide estimates based on observations beyond the seasonal cycle (mainly derived*

*from the Takahashi et al 2009 climatology). Therefore, we do believe that our observation based estimate provides new insight beyond the results from the RECCAP project, yet in turn we agree that our new estimates needs to be put in perspective with previous findings. Therefore, we will add a comparison between the ocean RECCAP results and the results from this study in the text.*

Section 5 overall:
- This section is also poorly organized. It reads as a listing of issues largely already mentioned prior. It needs to be rewritten to focus on the key findings of this work –
What are the take-home messages that the reader should be getting?

*We will reformulate the conclusions by focusing only on topics that we believe warrant the most attention in future research. These are listed in the response to the major comment #4*

Table 2: - should note that negative is from the atmosphere.

*Thanks, clarification will be added.*

- If the label is –GPP then GPP should be 108.29 not -108.29

*We agree, thanks.*

- A consistent number of significant figures should be used, unless the authors can justify the greater precision of the numbers with 5 significant figures (GPP) as opposed to those with only 2 or 3. This is important because it is uncertainty in GPP that drives most of the NCE uncertainty. The GPP numbers is clearly not actually known to 5 significant figures.

*We use two digits after the comma for all estimates (except CV because of the very small numbers). The uncertainties reported here are not used to calculate NCE uncertainties. Rather, the ensemble of NCE estimates (see response above explaining the uncertainty propagation) is used to estimate this uncertainty. In this way rounding errors do not propagate through the uncertainty estimation.*

- All numbers should have the same fontsize, or if the different sizes have a meaning, it should be noted

*Thanks, we will correct this. All font sizes should be equal.*

- The full decomposition of the "marine" should be noted in this table, so as to be consistent with Figure 1

*We list here only those fluxes that are used to estimate NCE (see also Eq. 1). Adding Estuaries and Shelves would be confusing because they don't enter the NCE calculation individually. We added them in Figure 1 for completeness. We will make this explicit in the title of the revised table.*

- The natural fluxes of the ocean need to be accounted for in a manner comparable to GPP and TER.

*As outlined above, we believe this is a misunderstanding. Our estimate comprises a combination of natural and anthropogenic fluxes, hence we do already account for natural „background" fluxes. Furthermore, as discussed above, we will only use directly upscaled NEP in the revised version to be more consistent between land and ocean.*

Figure 1
- The units on the 815 are presumably PgC. This should be noted explicitly on the figure or in the caption.

*Thank you, we will add the unit in the revised figure.*

- The ocean should have two arrows, one in and one out. The picture from this figure should be consistent with Figure 6.1 of IPCC in that both the ocean and the land have a large background, natural cycle on top of which the anthropogenic is superimposed.

*See above, all our estimates account for natural and background fluxes. Over land we will only use NEP in the revised version. Hence we will delete the arrows related to the gross fluxes over land.*

Figure 2
- The colorbar in panel a is mislabeled as "%"

*Thank you, the label should be gC m-2 yr-1. We will change this in the revised version.*

Figure 3
- The x-axis needs a label

*Thank you, the label should be latitude, we will change this in the revised version.*

Figure 4
- What are the circles? Presumably outliers? Clarify in caption.
- The regions indicated by each acronym should be noted in the caption.

*Thank you, yes, the circles are outliers, we will clarify this in the revised version and explain the acronyms.*

***Additional References:***

*Jacobson et al. A joint atmosphere-ocean inversion for surface fluxes of carbon dioxide: 1. methods and global-scale fluxes. Global Biogeochemical Cycles 21, 1, 2007.*

*van der Werf et al.: Global fire emissions estimates during 1997–2015, Earth Syst. Sci. Data Discuss., doi:10.5194/essd-2016-62, in review, 2017.*

*Figures:*

[Figure]

*Figure R1 (will be added in the revised manuscript)*
*Schematic explanation of the uncertainty propagation. Each spatiotemporal estimate of NCE is computed as the sum of randomly selected estimates of the 9 fluxes contributing to NCE (see Eq. 1, here denoted by $F_i$). For this study we compute 200 estimates of NCE. Uncertainties can now be assessed at different spatial scales by first aggregating all NCE estimates to the desired scale and then using the 200 members for uncertainty estimation.*

---

## Author Comment (AC2) · 27 Jan 2017

This paper puts together a wide range of spatially explicit bottom-up surface-atmosphere CO2 flux data sets aiming to reconcile the carbon budget from bottom-up estimation and the atmospheric CO2 growth rate. While this type of research is needed for improving our understanding of carbon cycle, this study has serious flaws in generating the data and is lack of validation and deep analysis of the combined data set. The language is vague in many places. At this stage, I don't recommend publishing the paper.

Major comments:
1. The added value of the new combined dataset is very limited.
The authors simply put different data streams together, and there is no effort trying to harmonize the data, even though some of the datasets do not cover the same time period, e.g., the crops cover 2005 to 2010.

*We strongly disagree with the impression that "there is no effort trying to harmonize… ". On the contrary, we have made major efforts to homogenize the various datasets comprising the current knowledge of spatiotemporally explicit, data-driven surface-atmosphere CO2 exchange. The chosen time period (2001-2010), spatial (1x1 degree) and temporal (monthly) resolution are a compromise arising from the availability of the different datasets. For several datasets, only one (annual mean) estimate for the chosen time period is available, including for Shelves, Estuaries, Rivers, Lakes, Wood harvest and the land use change flux (Eluc). All other datasets cover the entire time period with at least monthly time resolution and an original spatial resolution of 1x1 degree or finer such that resampling does not induce inconsistencies. Crop respiration data was extended backwards through linear extrapolation at each pixel. This is explained on p. 9 line 3.*

2. The paper compares the bottom-up estimations with the top-down inversion results (Figures 3 and 4, section 3.4), but it is lack of discussion about why these two approaches have different results, and which estimation is closer to reality.

*We will add a more in-depth discussion regarding these differences. We believe that our estimates overestimate carbon uptake in tropical land areas and carbon release in tundra regions. This may explain many of the differences visible in Figs. 3 and 4. For areas where the different estimates converge (mid-latitudes, Europe, Russia, South Asia, East Asia, Australia) we can state with some confidence that we know net carbon exchange. However, an overall judgment which estimates are closer to reality cannot be made given current knowledge.*

3. In section 3.4, it says that "both estimates agree well in the extratropics", but the figure 3d shows that the NCE-FF and atmospheric inversion results also have large differences in the NH high latitudes (between 60N and 75N), with the NCE-FF indicating a source to the atmosphere, while the atmospheric inversion indicating a weak sink.

*We agree with the reviewer and will add more discussion on this point in the revised version. At very high latitudes, in the tundra region, very few flux tower observations are available. Hence the FLUXCOM runs are not well constrained in those regions. In contrast to the tropics where this leads to an unrealistically large carbon sink, in the high latitudes the FLUXCOM runs show a strong source.*

4. Even though the latitudinal pattern of the inversion results follows a pattern similar to that of the aquatic fluxes (Figure 3c), there is no direct evidence indicating the propagation of the marine signal into continents during atmospheric inversion. I suggest removing the discussion on the pattern comparison between aquatic fluxes and atmospheric flux inversion results in section 3.4.

*We agree and will omit this discussion.*

5. Section 4.5 discusses the possible application of the combined dataset in model-data integration studies. It is an interesting idea. However, with such large uncertainties (with more than 10GtC disagreement with the atmospheric CO2 growth rate) in the combined dataset and a mixture of all different carbon flux components, it is not clear how such product can be used in carbon cycle data assimilation that focuses primarily on land carbon fluxes. What is the added value of using such data set compared to directly using flux tower observations? In addition, if such product were to be used as "observations" in a data assimilation system, a rigorous validation against independent observations is needed.

*Our main aim is to exploit the spatiotemporal explicitness of the NCE flux in tandem with the spatially explicit uncertainties for model-data fusion. Probably the most relevant application would be using these data at the regional scale, as one goal of the study is to pinpoint regions of small and large uncertainties in the NCE estimates. In some regions, uncertainties are so large that nearly no meaningful information on the mean NCE flux can be obtained with currently available observational networks and statistical approaches. This is, for example, the case for many tropical land regions. But, and we see this as a key advantage of our study, the included uncertainties clearly indicate the merit of such a data compilation, especially in contrast to flux tower observations: our study includes all the major fluxes, such as fire emissions, inland aquatic fluxes, tropical land use change estimates, and emissions related to harvested wood and crop products. This is much closer related (and more directly comparable) to the actual net carbon exchange fluxes as they are resolved by inversions (if fossil fuel emissions are omitted). All the datasets used in this study were validated individually against independent sources, and those studies are referenced in the respective sections. We don't know of any independent observation that can be used to validate the obtained NCE flux at such high spatial and temporal scale. An exception may be inversions and the regional aggregates obtained in the RECCAP synthesis, and we compare our estimates to RECCAP in the manuscript. We will include a similar comparison for the ocean regions in the revised version. At finer spatial and temporal scale, and in some regions, especially the tropics and northern high latitudes, independent trustworthy references are lacking.*

6. In section 3.2, it says that "13% of our runs we obtain a global C source that is consistent with the atmospheric growth rate", what are the spatial distributions of the fluxes from these 13% runs?

*As mentioned also in the response to reviewer 1, we have decided to omit this from the revised version. We base this on the following: (1) There is a large uncertainty related to missing fluxes (see page 16). (2) We cannot be sure that our assessment includes all relevant fluxes in all regions. (3) Constraining the NEP ensemble could thus be misleading because the constraint is highly uncertain. Furthermore, if we assume that relevant drivers are missing in the set of predictors in FLUXCOM (e.g. forest age, see Sect. 4.1), all members in FLUXCOM are biased in the same way (all members use the same set of predictors). Constraining the ensemble thus cannot provide us with new insights into the processes that need further investigation or drivers that are missing in the set of predictors.*

7. L26 (P14): what is the distribution of the different age classes of forests in FLUXNET? Is there solid evidence showing that the year and regrowing forests are overrepresented in FLUXNET?

*So far this is only a hypothesis and it has not been shown. This hypothesis is a strong candidate in explaining the overestimation of the carbon sink in the tropics. Future research has to demonstrate whether these hypotheses are valid. We will make this clearer in the revised version.*

8. Section 3.5 discusses seasonal cycle and monthly variability. It would be helpful to put this discussion in perspective, e.g., comparing to other independent estimations, so that the readers would know the credibility of this result. It is not clear what are the latitude ranges for the NH and SH in Figure 6.

*The ranges are 90 S-0 for SH and 0-90 N for NH. We will include a comparison with inversions for the seasonal cycle, which are the only independent estimates based on observational data. As annual variability is already compared to inversions demonstrating large discrepancies (Figure 5) we refrain from comparing monthly variability.*

9. Line 6 (p15), what is the basis for the 50% uncertainty?

*Here we refer to a recent paper on global fire emissions by van der Werf et al, currently in discussion for Earth System Science Data (van der Werf et al., 2017). Estimating uncertainties in fire emission estimates is notoriously difficult. Assuming 50% uncertainty for estimated fire emissions is a best guess assessment, and better quantifying this uncertainty requires an assessment of the burned area estimates as well as new field data on fuel consumption and emission factors.*

Minor comments
1. In the abstract, "would require an offsetting surface C source of 4.27 +- 0.10 PgC/yr", should the offset be 4.27 + 6.07 PgC/yr in order to have 4.27 PgC atmospheric $CO_2$ growth rate?

*Yes, that is correct. We will reformulate this sentence to make this clearer.*

2. Line 13-16 (p3), it is not clear what the "background $CO_2$ fluxes" means.

*With background fluxes we mean the fluxes before human disturbance (i.e., before the large increase in fossil fuel emissions). Those are not included in the estimates of the Global Carbon Project which only discusses the human perturbation. In the revised version we will avoid the term to avoid confusion.*

3. Line 23 (p4), "goal of this study the" should be "goal of this study to"

*Thank you.*

4. Line 16 (p6): what is "schused"?

*Should be "used", will be changed in the revised version.*

5. Line 21 (p14): What does the "relevant drivers" refer to? Be more specific.

*Additional drivers relevant for upscaling NEP could be, for instance, the disturbance history (e.g. time since the last disturbance) or, closely related, forest age. This is mentioned a few lines higher up. We will be more specific here and put the most likely missing drivers.*

6. Line 29 (p14): what does the "global driver" refer to?

*By this we refer to the fact that there is no global map of forest age, which could be used as an additional driver for upscaling NEP (see also comment above). We will clarify this in the revised version.*

**Additional References:**

*van der Werf et al.: Global fire emissions estimates during 1997–2015, Earth Syst. Sci. Data Discuss., doi:10.5194/essd-2016-62, in review, 2017.*

---

## Author Response (AR1)

***Reviewer 1***

*We would like to thank reviewer #1 for the detailed review of our manuscript and the thoughtful suggestions that will help to improve our manuscript. In the following, we will answer each of the reviewers comment.*

Zscheischler et al. pull together a variety of surface to air CO2 flux estimates and ask the question "Do these add up to a globally balanced budget?" This is a worthwhile effort, and the authors are using state of the art estimates. As alluded to in the text, the primary goal of this work is to create a combined data product that can be used as input to future data assimilation efforts. Unfortunately, there are vital errors the analysis. Large annual cycles of CO2 flux are taken into account for land, but entirely ignored for the ocean.

*We believe this is a misunderstanding. We do consider annual cycles of all variables – where these were available. In particular, we consider annual cycles over the main fluxes over land* and *ocean. This is also highlighted in Table 1, where the temporal resolution of each dataset is listed. The ocean fluxes are spatiotemporally explicit at monthly time scale and were estimated by Rödenbeck et al (2014) and Landschützer et al (2014) as explained in section 2.2.1.*
*State-of-the-art observation-based estimates of shelf areas and inland waters are however still missing the seasonal representation. This kind of gap analysis is exactly what we intend to do here: demonstrate where we currently miss information to achieve a comprehensive and purely data driven description of the surface-atmosphere CO2 exchange. We believe that this is the best way forward to improve our future understanding and fill these knowledge gaps.*

The authors suggest that they are looking at the full "background" of natural CO2 fluxes, but only consider the anthropogenic perturbation in the ocean. To be correct and consistent with the statements of a full accounting for natural background fluxes, Table 1 and Figure 1 should have large fluxes in the ocean that are of the order of the GPP and TER on land. Furthermore, just assembling these data-based estimates into one with global coverage is not sufficient for publication.

*To the best of our knowledge this is the first critical appraisal of data driven estimates of surface-atmosphere CO2 fluxes, which may be relevant for wide community working in C cycle science. For the ocean we provide estimates of the contemporary carbon fluxes, i.e. a combination of natural and anthropogenic fluxes. Based on surface ocean pCO2 observations the natural and anthropogenic components cannot be separated (see also below).*

The analysis here is too thin, and the findings are poorly presented. Based on how the independent products have been produced, no one should expect that they would add up to a balanced budget – this finding is no surprise.

*Indeed, we don't expect the readers to be surprised that individual components do not close the carbon budget. However, our spatially explicit description of data uncertainty and budget mismatch is of key importance to guide future research efforts. Alternatively, it is crucial to show where the independent products do not add up and where the largest observational knowledge gaps and uncertainties are. Our effort is the first to do this systematically across all components of the surface-atmosphere $CO_2$ exchange global carbon cycle.*

The authors do not do enough to explain what are the major sources of the uncertainty, nor do they do enough to make it clear how they estimate this uncertainty.

*We apologize if we did not achieve a sufficient description of how we estimate uncertainty. To improve the presentation and to make it more clear how we derive our uncertainties, we have made substantial revisions of the text explaining the uncertainty propagation and also included a visual description of the work flow (see new Figure 1 in the revised manuscript and explanation below).*

They need to do a lot more with the products that they have before this manuscript is acceptable for publication.

*We agree that the data we present here would allow much more analysis and we would encourage the community to use the datasets and add additional analysis. Overall, however, we would like to reemphasize that the main aim of this study is not to simply check whether data-driven surface-atmosphere $CO_2$ fluxes add up to a balanced budget. Given the difficulties of guaranteeing a consistent and contiguous global C monitoring system, this simply cannot be expected. And the current data-driven knowledge about many of the relevant fluxes cannot compensate this. But – and we find this an important contribution – we provide global spatiotemporal estimates of the net carbon flux combining a variety of heterogeneous datasets and consistently propagate their uncertainties. We have identified regions of high and low uncertainty, guiding new monitoring campaigns and novel scientific approaches to reduce specific uncertainties. Our NCE estimates specify contemporary fluxes over the whole Earth surface, thus including background and anthropogenic fluxes, as explained below. Note also that this paper is not considered a normal Research paper but a Synthesis (see manuscript types of Biogeosciences).*

Major Comments
1. The authors indicate that their goal is to not just address anthropogenic carbon uptake, but to also address the background carbon fluxes (Page 3). Yet their methodology is inconsistent across land vs ocean in this respect. While on land, they separate GPP uptake of CO2 from TER efflux, they completely ignore the comparable cycle in the ocean. See Figure 6.1 of Ciais et al. 2013 (IPCC WG1, Chapter 6) where it is clear that the naturally occurring cycle in the ocean creates an exchange flux of 80 PgC/yr out of the ocean and 78.4 PgC/yr into the ocean; this is comparable in magnitude to the GPP and TER (+-100 PgC/yr), but the authors here simply ignore these ocean fluxes by only presenting their sum. They also appear to ignore these large fluxes in their assessment of uncertainty (though detail on how uncertainty is accounted for is so thin that it is hard for the reviewer to be sure on this point). The full background cycle in the ocean must be included in this analysis must be remedied in this analysis.

*The reviewer is correct that we have only displayed aquatic net fluxes (not only for the open ocean but throughout the whole aquatic system). This is a result of data availability. We also concur that the gross fluxes may have individually larger uncertainties attached to them. We do, however, disagree, that we „ignore" these fluxes or their uncertainty. The uncertainty of the net flux presented in this study is comparable to other uncertainty estimates such as the IPCC report or the Global Carbon Budget (e.g. Le Quéré et al 2015). The reason for the smaller net uncertainty stems from the correlation between air-sea and sea-air fluxes and their uncertainty. The largest source of uncertainty between individual flux elements (sea-air or air-sea) stem from the gas transfer formulation (see also answer to a more specific comment below), i.e., a systematic source of uncertainty which likewise impacts fluxes in both directions leading to a much smaller net flux difference and attached uncertainty. This is also stated in the caption of the IPCC mentioned by the reviewer: "Individual gross [air–sea exchange] fluxes and their changes since the beginning of the Industrial Era have typical uncertainties of more than 20%, while their differences (Net land flux and Net ocean flux in the figure) are determined from independent measurements with a much higher accuracy (see Section 6.3).*

*Therefore, to achieve an overall balance, the values of the more uncertain gross fluxes have been adjusted so that their difference matches the Net land flux and Net ocean flux estimates."*

*Over land we have used TER and GPP because these fluxes are available at the spatiotemporal grid which we required. This is not the case for the ocean. However, we agree with the reviewer that this is inconsistent. In the revised version, we therefore only used the directly upscaled NEP product from FLUXCOM, which reduces the sample size of the NEP estimates from 16 to 8 (the uncertainty related to the flux separation (split of NEP into GPP and TER) is dropped, as it is not relevant for the uncertainty estimation of the net fluxes). The uncertainty in the upscaled NEP product is 2.1PgC/yr (compared to 3.4PgC/yr when using TER-GPP), which is still much larger than the uncertainty of the net flux over the ocean (0.15PgC/yr). Using directly upscaled NEP leads to similar global mean estimates than using TER-GPP (the difference is <0.7PgC/yr, i.e., <5%).*

*In response to the reviewer's comments we have also added above explanation regarding net uncertainty of ocean fluxes in the text. We do however avoid the term „background fluxes" as this term can be easily confused with „natural fluxes", whereas we cannot separate natural and anthropogenic components. All our flux estimates are the aggregates of natural fluxes and anthropogenic disturbance (i.e. the contemporary flux). We have made this explicit in the introduction by writing "Unlike the GCP global budget of anthropogenic $CO_2$, we consider here the full contemporary exchange of surface-atmosphere $CO_2$ fluxes."*
*More details regarding the ocean fluxes are added below in direct response to an additional comment by this referee.*

2. A coherent explanation for the large imbalance in the final "budget" is never presented, instead the reader is left is a laundry list (e.g. page 16) of possibilities and no clarity of what the authors have identified as the likely most important uncertainties. It seems quite likely that the large GPP and TER fluxes, or the comparable ocean fluxes, are biased high or low. Their uncertainties are the only ones on the same order as the NCE uncertainty. This issue is even more obvious at the regional scale. This issue should be more directly addressed.

*In the revised version we have more clearly emphasized and discussed the most likely reasons for this imbalance. In our opinion it is most likely a combination of*
> *i)       a bias in NEP, most probably in the tropics where only very few eddy covariance sites lead to a weak observational constraint (section 4.1).*
> *ii)      missing sources (as listed in section 4.2), especially emissions from wetlands and VOCs.*

*This was also added to the Conclusion section.*

a. One clear place to do this would be on Page 12, where it is stated that in 13% of their runs, the global C source is consistent with the atmospheric growth rate. If this finding is meaningful, these 13% of runs need to be analyzed and presented clearly so that the reader can understand what is different about them. A simple explanation for the uncertainty in the budget could be that GPP is overestimated by 10%, and if all of these 13% of runs have GPP on the low side, then it would be useful to identify such a pattern.

*In the original version, 13% of the runs were consistent with the atmospheric growth rate only when assuming neutral exchange in tropical forests. In the revised version we use directly upscaled NEP and none of the runs is consistent with the atmospheric growth rate. This is related to the fact that the uncertainty related to flux separation (dividing NEP into TER and GPP) is not included anymore,*

*which is more consistent because NEP is measured directly by eddy-covariance towers. In addition the large amount of missing fluxes may also inhibit such a constraining exercise, since the magnitude of these omitted fluxes would largely determine which runs would get selected.*

3. There are many inconsistencies in the data products used here. For example, for the ocean flux the parameterization of gas exchange is Wanninkhof (1992) with ERA-interim winds, but for the shelf it is Wanninkhof et al. (2013) with CCMP winds. These differences could make a significant difference to the ultimate fluxes even though based on the same pCO2 database. On the one hand, this reviewer recognizes that these differences are due to choices made by the providers of these previously-published flux products, and cannot be easily changed by these authors. Nevertheless, some evaluation of these effects should be performed. One possibility for such evaluation could be in the overlap regions of the three products that go into the merged Marine flux field.

*We agree that there are inconsistencies between the data products. As noted by the referee, it is not the aim of the study to re-calculate estimates, but rather bring together existing knowledge. By including inconsistencies between currently available state-of-the-art estimates we also implicitly sample uncertainty. Nevertheless, we have tried to account for many inconsistencies, i.e., we calculated flux estimates at the same spatial and temporal resolution, we have unified the uncertainty calculation procedure, we have recalculated overlap areas to avoid double accounting, etc, but as rightfully noted by the referee, there are still some other sources we have not accounted for. The referee highlights the gas exchange formulation as an example and we concur that this is a factor that has a significant impact on the air-sea exchange of CO2. However, as explained above, the net effect of the gas flux formulation is of lesser importance when the net flux is considered. We would also like to note that while the open ocean estimates use the formulation of Wanninkhof et al 1992, i.e., a quadratic dependency between gas flux and wind speed at 10m, they use more recent gas transfer coefficients (Rödenbeck et al and Landschützer et al scale their estimates to match a mean transfer velocity of 16cm/hr as suggested by Wanninkhof et al 2013). However, we concur with the reviewer that there is an additional uncertainty related to the transfer. In this way, uncertainty in ocean estimates is probably underestimated. This is, however, also true for land based estimates. E.g. in FLUXOM, all models use the same set as predictors. We welcome the suggestion of the reviewer to use the overlap area for testing, however, this overlap area is very local and is biased towards coastal zones. We have added a paragraph at the beginning of section 4.5 discussing this probable underestimation of uncertainty:*

*"Our uncertainty estimates of ocean and land C exchange likely underestimate the true uncertainty. In particular, Landschützer et al. (2014) estimated that the choice of transfer formulation and the pCO$_2$ mapping mismatch (also including other relationships than quadratic) lead to an uncertainty of 37% for the global average over sea-air exchange between 1998-2011, with the majority of this uncertainty stemming from the gas transfer formulation. Furthermore, the uncertainty of NEP is likely underestimated because all upscaling methods in FLUXCOM use the same set of driver data (Tramontana et al., 2016). Hence, the uncertainty estimates only cover the uncertainty related to the upscaling method but does not cover uncertainties related to the selection of optimal drivers or observational uncertainty of the drivers themselves."*

4. The text is difficult to follow, particularly in the discussion and conclusion sections. These sections read as a list of possibilities, without clarity as to what is really important. The authors need to do more to provide this needed clarity.

*We have rewritten large parts of the text to achieve a better readability. We have reformulated the*

*discussion and conclusion sections to emphasize and discuss our main results better. In particular, we have added an introductory overview paragraph in the discussion. We have also rearranged the subsections of the discussion according to their relevance. Our conclusions now focus on the following main results:*

  *i)  Current spatiotemporally explicit observation-driven estimates of surface-atmosphere CO2 exchange are not constrained well enough to close the carbon budget at the global scale.*

  *ii)  The most likely candidate for inducing the mismatch between data-driven surface-atmosphere CO2 exchange and the atmospheric CO2 growth rate is land NEP, in particular tropical NEP estimates appear to be strongly overestimated (too large sink). Understanding this bias will help designing better upscaling approaches (e.g. by including currently missing relevant drivers) and pinpointing variables that need to be (better) monitored in the future.*

  *iii)  Regionally, those estimates are partly well constrained and may be used for model-data integration studies and validation of models. These regions include Europe, Russia, South Asia, East Asia, Australia and most oceanic regions. Better constraining C fluxes in regions with currently high uncertainties should be a priority of future research.*

Minor comments
- Line 15 "limitations."

*Thanks.*

- Line 24 Which regions have the large net sink? The authors specify several regions with flux to the atmosphere, then the sum of all is large and negative, presumably due to the tropics. The reader should be able to better understand where this large negative is coming from geographically based on the abstract.

*The large C is over found over most of the tropical land, i.e., Amazon, Congo and Indonesia. We have revised this section in the abstract as follows: "Our NCE estimates give a likely too large $CO_2$ sink in tropical areas such as the Amazon, Congo and Indonesia. Overall, and because of the over-estimated $CO_2$ uptake in tropical lands, our global bottom-up NCE amounts to a net sink of -5.4±2.0 PgC/yr. By contrast, the accurately measured mean atmospheric growth rate of $CO_2$ over 2001-2010 indicates that the true value of NCE is a net $CO_2$ source of 4.3±0.1 PgC/yr."*

- Line 15: "background CO2 fluxes over land and ocean," This analysis only accounts for background fluxes in the land, not in the ocean. Instead this analysis suggests there is no net background ocean flux, only the anthropogenic residual! In contrast to Figure 6.1 of IPCC WG1 (Ciais et al. 2013), this analysis ignores background, natural ocean exchanges. This is a major error that must be remedied.

*We believe that this is a misunderstanding. The provided flux estimate is neither natural nor anthropogenic but the contemporary flux, i.e., a combination of both natural and anthropogenic. All our estimates, both one land and over the ocean, represent this contemporaneous flux. We have made this clearer by revising the sentence about the GCP: "The budget of the GCP focuses on annual values integrated at the global scale. An important point is that the GCP budget quantifies solely the anthropogenic perturbation of $CO_2$ fluxes, i.e., it provides information about the fate of anthropogenic $CO_2$ emissions in natural reservoirs" Furthermore, later on in the introduction we now state "Unlike*

*the GCP global budget of anthropogenic $CO_2$, we consider here the full contemporary exchange of surface-atmosphere $CO_2$ fluxes."*

- Line 2 What is the meaning of "resampled". Is this averaging of all points in a 1x1 grid? If data are at coarser resolution, what is done? Please be more specific. Show that global mean values of the variables considered are conserved by this method.

*All datasets have at least 1x1 degree resolution (1x1 degree or finer), such that resampling here means averaging to this coarser resolution. All global means are conserved by this averaging (by taking the land-ocean masks into account). We have rewritten this section as:*
*"Each dataset was aggregated to 1 x 1 degree spatial resolution. All datasets have an original spatial resolution of at least 1 x 1 degree such that no information was lost through re-gridding. The temporal resolution is monthly. For datasets that were only available at yearly time scale or once over the complete time period (Table 1), we distributed fluxes evenly across all months."*

- Line 25 "For NCE estimates, we randomly combined all datasets, using a single realization of each flux, to generate an estimate of NCE." What is the meaning of "randomly combined all datasets"? How is this random if all datasets are used? If the "random combination" applies only to the 2 fluxes (ocean and LUC) that have multiple sources according to Table 1, then the result here is an incomplete estimate of uncertainty. More explanation is needed here so that the reader can have confidence in the uncertainty estimate being made.

*The random combination applies always to all fluxes contributing to NCE. That is, we create multiple estimates of NCE by summing up different random combinations of the source datasets on the right side of Eq. 1. Uncertainty is then estimated based on the newly generated NCE ensemble. We have rewritten section 2.1 to better explain how uncertainties were propagated. See also the response to the comment below.*

- Overall it is hard to understand how the uncertainty is propagated. Bits and pieces are mentioned under each of the flux products below, but a coherent picture is not made clear. Perhaps this lack of clarity could be partially remedied with a schematic figure that clarifies how many different realizations of each flux and how the sampling across them is performed.

*We have introduced a schematic figure to better explain how the uncertainties are propagated, as the reviewer suggested (new Figure 1). Each of the 200 NCE ensemble members consists of the sum of randomly selected members of the fluxes contributing to NCE (see Eq. 1). In principle, we could create 10\*10\*50\*8\*10\*2=800.000 different NCE estimates by combining all the available members (see the #Runs shown in Table 1). We limit ourselves to 200 NCE estimates as a representative sample for the whole distribution due to the prohibitive computational expense of running all 800.000 combinations. All NCE uncertainty estimates are then derived from these 200 runs. This approach implicitly contains information on the spatiotemporal uncertainty structure of the NCE estimates (i.e., the error covariance matrix). In this way, regional or continental NCE uncertainties can be computed by aggregating each of the 200 NCE estimates over the desired region, automatically taking correlated errors into account. This was for example done for Figures 5 and 6.*

- Line 16 "schused" is a typo

*Thanks, should be "used" and has been changed.*

- Line 15 If the same FLUXCOM product is being used to separate the gpp and ter, the ocean fluxes out vs in should also be separated (Figure 6.1, Ciais et al. 2013). It is inconsistent to take different approaches with land vs ocean, and skews the reader impression of the magnitude of local fluxes and their uncertainty.

*As explained above, in the revised version we have only used the directly upscaled NEP product over land to be consistent and to focus the attention on the uncertainties that are related to the net fluxes only.*

- Line 8: A reference for EDGAR is likely warranted.

*As suggested by the EDGAR-Terms of use, we have acknowledged the EDGAR data providers in the revised version.*

- Line 24 "Not all inversions were available till 2010." What is done if this is the case?

*The mean and uncertainty for each year is taken over all available inversions for that year. We have specified this in the revised version in the section on inversions.*

- Line 6: That this imbalance is not real, but an artefact of the uncertainty of the data should be made explicitly clear here; not just left for section 4.

*We agree with the reviewer that this statement may be misunderstood. We report the mismatch which is obtained when combining all currently available spatiotemporal data-driven surface-atmosphere fluxes. This statement thus highlights that our current knowledge on C fluxes is not sufficient to close the C budget in this way (i.e., leaving out process-based models). We have added the following comment to this mismatch: "Thus, there is a large mismatch with our NCE, which over-estimates the $CO_2$ sink at the surface by $9.7\pm2.0$ PgC / year. This highlights that our observation-based NCE is biased towards a too large sink."*

- Line 12: "whereas in fact many errors might be correlated as this is clearly the case for GPP and TER." The same statement will almost certainly be appropriate in the case of uncertainty in ocean fluxes, once they are appropriately accounted for.

*This is true (as also argued above). However, in contrast to what the reviewer is implying here, correlated errors lead to smaller uncertainties. As stated above, the uncertainties in the gross fluxes over the ocean are indeed highly correlated, resulting in much smaller uncertainties for the net fluxes (see also the caption of the IPCC figure mentioned by the reviewer). By using the ocean net flux estimates for NCE, these correlated uncertainties are automatically accounted for. As stated before, in the revised version we only used directly upscaled NEP over land. Hence, the comparison of the uncertainties is now consistent and we could omit this statement.*

- Line 22: "Due to the small contribution of the oceans, absolute uncertainties are barely discernible." Comment: This will probably be different once out vs in is considered separately.

*As outlined in our response to the reviewer's major comment above, the net sink uncertainty is in fact much smaller than air-sea and sea-air flux uncertainty, due to the correlation between the uncertainties between the individual components. Using NEP over land (instead of TER-GPP) in the revised manuscript decreases this difference slightly (see Figure 3 in the revised manuscript).*

- Line 14 "We use the land cover map of 2005 from the European Space Agency (http://www.esa-landcover-cci.org/) to identify tropical forests (all pixels where broadleaved evergreen trees dominate). " Why use satellite product, when FLUXCOM model is what your estimate is based on. The FLUXCOM land cover product should be used.

*Thank you for this comment, we have now used the land cover map of FLUXCOM.*

- Line 3-11: This section overstates the level of agreement with Ciais et al. (in revision). It suggests that Figure 3 illustrates "good agreement" except for a few regions, but when one looks closely, only 5 land regions agree, including Australia that is basically zero, while 4 do not. Overstatement is exemplified by this sentence "Given that Ciais et al. (revision) rely on an independent method, this demonstrates that a good understanding of net C fluxes exists for non-tropical areas, North America excluded." This section should be written more carefully to acknowledge that lack of agreement is as common as agreement.

*We agree with the reviewer and have formulated this more carefully in the revised version, better highlighting the large uncertainties and biases. The revised sentences read "Given that Ciais et al. (revision) rely on an independent method, this demonstrates that a relatively good understanding and observational coverage of net C fluxes exists for EU, RU, AU, and EA to some extent. It is somewhat surprising that both approaches largely differ over North America, where good observational coverage for instance through eddy-covariance towers exist."*

- Line 6: NEP should be defined again as its not been defined for many pages.

*Thanks, NEP is now used repeatedly as one of the key variables in the revised version such that there is no need to define it again.*

- Line 23-end: This reads as it may be one hypothesis of many. Or is it a leading one? The reader needs the authors to be more clear.

*This is the leading one. We have emphasized this more in the revised version by starting the sentence with "We suspect that…". To test this hypothesis rigorously, the complete upscaling procedure needs to be redone with including a forest age map as a predictor, which should be done in future research.*

- Line 8: "often not too far off but given that different top-down studies using different atmospheric models provide conflicting information on the adjustments needed to align modelled concentrations with measured ones, this information cannot be used to provide clear uncertainty ranges." This is not understandable to someone doesn't work with these models or know this jargon.

*We have rephrased the section regarding uncertainties in fire emission referring to a recent paper on global fire emissions by van der Werf et al, currently in discussion for Earth System Science Data (van der Werf et al., 2017). Here it is stated that assuming 50% uncertainty overall is a best guess assessment, and better quantifying this uncertainty requires an assessment of the burned area estimates as well as new field data on fuel consumption and emission factors. We cannot, however, propagate this uncertainty into the NCE estimates as this would require spatiotemporal error covariance matrices.*

- Line 13 "To better constrain C exchange on a monthly basis, however, the seasonal cycles of those fluxes would be necessary." Comment: Awkward phrasing.

*We have rephrased this statement as "Estimates of the seasonal variation in these fluxes are necessary to better constrain seasonal variations in NCE."*

- Line 23 "at similar latitudes."

*Thanks.*

- The degree to which the unaccounted fluxes (list) could be large enough to impact the global "budget" should be discussed. Each of these fluxes should be quantified to the best degree possible so as to put in context with the overall budget. The laundry list approach is not helpful to the reader, particularly when so many of the proposed fluxes are left entirely unquantified, and the authors do not discuss the list at all after it is presented. What is the reader to meant to conclude?

*Thank you for this comment. We have provided estimates for those fluxes, which have been quantified in the past. The remaining fluxes can be assumed to be rather minor, though little is currently known. We have discussed to what degree the missing fluxes affect the obtained mismatch by adding the following paragraph at the end of this section: "All known missing fluxes add up to an additional C release of about 4 PgC / yr. Although substantial, they do not cover the mismatch of more than 9 PgC / yr by far (Sect. 3.1). However, they would suffice to close the budget if tropical forests are assumed to be C neutral (tropical forests are responsible for a net C sink of about 5 PgC / year, Sect. 3.2). This significant amount of missing fluxes prohibits constraining FLUXCOM runs with all the remaining fluxes. In other words, we cannot be certain of the bias in upscaled NEP as long as the major fluxes are not quantified in a spatially explicit manner. Emissions from VOCs and wetlands should thus receive particular attention if a consistent spatiotemporal picture of vertical $CO_2$ exchange is to be obtained."*

- Line 14: Regions such as the North Atlantic (Schuster et al. 2013, Biogeosciences) should also be noted as having large uncertainty at seasonal timescales and beyond.

*The RECCAP initiative has shown that the largest uncertainties are in the southern hemisphere, i.e., in Schuster et al 2013, the South Atlantic has shown much less agreement between methods than the North Atlantic when low frequency signals – such as IAV and trends – are considered, whereas, methods generally agree seasonally where the seasonal cycle is dominated by the temperature variability, i.e., subtropics. In general, the ocean RECCAP studies (Sarma et al, Ishii et al, Schuster et al, Lenton et al and Wanninkhof et al) have shown regionally substantial differences between methods, however, few of these papers provide estimates based on observations beyond the seasonal cycle*

*(mainly derived from the Takahashi et al 2009 climatology). Therefore, we do believe that our observation-based estimate provides new insight beyond the results from the RECCAP project, yet in turn we agree that our new estimates need to be put in perspective with previous findings. Therefore, we have added a comparison between the ocean RECCAP results and the results from this study. The new figure 6 shows that estimates of mean annual C exchange are consistent for most ocean regions, including the North Atlantic (see also the new section 3.3.2). Only in the Southern Ocean (SO)are the estimates are substantially different. Overall, our estimates have smaller uncertainty ranges, which is to be expected as the RECCAP studies include many more approaches (including process-based models, atmospheric and ocean inversions) in their estimates.*

Section 5 overall:
- This section is also poorly organized. It reads as a listing of issues largely already mentioned prior. It needs to be rewritten to focus on the key findings of this work –
What are the take-home messages that the reader should be getting?

*We have rewritten the conclusions by focusing only on topics that we believe warrant the most attention in future research. These topics are highlighted as a response to major comment #4 (see above).*

Table 2: - should note that negative is from the atmosphere.

*Thanks, clarification has been added.*

- If the label is –GPP then GPP should be 108.29 not -108.29

*GPP is always positive. Hence, -GPP should receive a negative number. In the revised version we use NEP instead of GPP and TER. Also here, –NEP gets negative numbers to maintain consistency with all fluxes in the table.*

- A consistent number of significant figures should be used, unless the authors can justify the greater precision of the numbers with 5 significant figures (GPP) as opposed to those with only 2 or 3. This is important because it is uncertainty in GPP that drives most of the NCE uncertainty. The GPP numbers is clearly not actually known to 5 significant figures.

*We use two digits after the comma for all estimates. The uncertainties reported here are not used to calculate NCE uncertainties. Rather, the ensemble of NCE estimates (see response above explaining the uncertainty propagation) is used to estimate this uncertainty. In this way, rounding errors do not propagate through the uncertainty estimation.*

- All numbers should have the same fontsize, or if the different sizes have a meaning, it should be noted

*Thanks, we have corrected this. All font sizes should be equal.*

- The full decomposition of the "marine" should be noted in this table, so as to be consistent with Figure 1

*We list here only those fluxes that are used to estimate NCE (see also Eq. 1). Adding Estuaries and Shelves would be confusing because they don't enter the NCE calculation individually. We added them in Figure 1 (now Figure 2) for completeness. We have made this explicit in the title of the revised*

*table.*

- The natural fluxes of the ocean need to be accounted for in a manner comparable to GPP and TER.

*As outlined above, we believe this is a misunderstanding. Our estimate comprises a combination of natural and anthropogenic fluxes, hence we do already account for natural „background" fluxes. Furthermore, as discussed above, we have used directly upscaled NEP in the revised version to be more consistent between land and ocean.*

Figure 1
- The units on the 815 are presumably PgC. This should be noted explicitly on the figure or in the caption.

*We have omitted the amount of carbon in the atmosphere to be more consistent overall and only show fluxes/changes in atmospheric C.*

- The ocean should have two arrows, one in and one out. The picture from this figure should be consistent with Figure 6.1 of IPCC in that both the ocean and the land have a large background, natural cycle on top of which the anthropogenic is superimposed.

*See above, all our estimates account for natural and background fluxes. Over land we have only used NEP in the revised version. Hence we have deleted the arrows related to the gross fluxes over land.*

Figure 2
- The colorbar in panel a is mislabeled as "%"

*Thank you, the label should be gC m-2 yr-1. Has been changed.*

Figure 3
- The x-axis needs a label

*Thank you, we have added the label "Latitude".*

Figure 4
- What are the circles? Presumably outliers? Clarify in caption.
- The regions indicated by each acronym should be noted in the caption.

*Thank you, yes, the circles are outliers. Due to the use of NEP in the revised version, no outliers appear anymore. We have also explained the acronyms.*

**Reviewer 2**
This paper puts together a wide range of spatially explicit bottom-up surface-atmosphere CO2 flux data sets aiming to reconcile the carbon budget from bottom-up estimation and the atmospheric CO2 growth rate. While this type of research is needed for improving our understanding of carbon cycle, this study has serious flaws in generating the data and is lack of validation and deep analysis of the combined data set. The language is vague in many places. At this stage, I don't recommend publishing the paper.

Major comments:

1. The added value of the new combined dataset is very limited.
The authors simply put different data streams together, and there is no effort trying to harmonize the data, even though some of the datasets do not cover the same time period, e.g., the crops cover 2005 to 2010.

*We strongly disagree with the impression that "there is no effort trying to harmonize… ". On the contrary, we have made major efforts to homogenize the various datasets comprising the current knowledge of spatiotemporally explicit, data-driven surface-atmosphere CO2 exchange. The chosen time period (2001-2010), spatial (1x1 degree) and temporal (monthly) resolution are a compromise arising from the availability of the different datasets. For several datasets, only one (annual mean) estimate for the chosen time period is available, including for Shelves, Estuaries, Rivers, Lakes, Wood harvest and the land use change flux (Eluc). All other datasets cover the entire time period with at least monthly time resolution and an original spatial resolution of 1x1 degree or finer such that resampling does not induce inconsistencies. Crop respiration data was extended backwards through linear extrapolation at each pixel. This is explained on p. 9 line 3. We have highlighted these aspects better at the beginning of section 2 an also introduced a schematic figure to better explain the consistent propagation of uncertainties. Note also that this paper is not considered a normal Research paper but a Synthesis (i.e., the goal is to "summarize the status of knowledge and outline future directions of research", see manuscript types of Biogeosciences).*

2. The paper compares the bottom-up estimations with the top-down inversion results (Figures 3 and 4, section 3.4), but it is lack of discussion about why these two approaches have different results, and which estimation is closer to reality.

*We have added a more in-depth discussion regarding these differences (see revised section 4). We believe that our estimates overestimate carbon uptake in tropical land areas and carbon release in tundra regions. This may explain many of the differences visible in Figs. 4 and 5. For areas where the different estimates converge (mid-latitudes, Europe, Russia, South Asia, East Asia, Australia) we can state with some confidence that we know net carbon exchange. With respect to regions where estimates diverge strongly, an overall judgment which estimates are closer to reality cannot be made given current knowledge.*

3. In section 3.4, it says that "both estimates agree well in the extratropics", but the figure 3d shows that the NCE-FF and atmospheric inversion results also have large differences in the NH high latitudes (between 60N and 75N), with the NCE-FF indicating a source to the atmosphere, while the atmospheric inversion indicating a weak sink.

*We agree with the reviewer and have added more discussion on this point in the revised version. At very high latitudes, in the tundra region, very few flux tower observations are available. Hence the FLUXCOM runs are not well constrained in those regions. In contrast to the tropics where this leads to an unrealistically large carbon sink, in the high latitudes the FLUXCOM runs show a strong source. We have added in section 4.1 "In addition to the above difficulties, some regions are undersampled by eddy-covariance towers and thus NEP is not well constrained. This is the case for tropical forests and the northern high latitudes. In the tropics, undersampling leads to a large overestimation of net $CO_2$ uptake in comparison to inversion and forest inventories (Peylin et al., 2013; Pan et al., 2011) whereas in the high latitudes it leads to a comparably large $CO_2$ release (Figure 4)."*

4. Even though the latitudinal pattern of the inversion results follows a pattern similar to that of the

aquatic fluxes (Figure 3c), there is no direct evidence indicating the propagation of the marine signal into continents during atmospheric inversion. I suggest removing the discussion on the pattern comparison between aquatic fluxes and atmospheric flux inversion results in section 3.4.

*We agree and have omitted this discussion.*

5. Section 4.5 discusses the possible application of the combined dataset in model-data integration studies. It is an interesting idea. However, with such large uncertainties (with more than 10GtC disagreement with the atmospheric CO2 growth rate) in the combined dataset and a mixture of all different carbon flux components, it is not clear how such product can be used in carbon cycle data assimilation that focuses primarily on land carbon fluxes. What is the added value of using such data set compared to directly using flux tower observations? In addition, if such product were to be used as "observations" in a data assimilation system, a rigorous validation against independent observations is needed.

*Our main aim is to exploit the explicit spatiotemporal nature of the NCE flux in tandem with the spatially explicit uncertainties for model-data fusion. The most relevant application would be using these data at the regional scale, as one goal of the study is to pinpoint regions of small and large uncertainties in the NCE estimates. In some regions, uncertainties are so large that nearly no meaningful information on the mean NCE flux can be obtained with currently available observational networks and statistical approaches. This is, for example, the case for many tropical land regions. But, and we see this as a key advantage of our study, the included uncertainties clearly indicate the merit of such a data compilation, especially in contrast to flux tower observations: our study includes all the major fluxes, such as fire emissions, inland aquatic fluxes, tropical land use change estimates, and emissions related to harvested wood and crop products. This is much closer related (and more directly comparable) to the actual net carbon exchange fluxes as they are resolved by inversions (if fossil fuel emissions are omitted). All the datasets used in this study have been validated individually against independent sources, and those studies are referenced in the respective sections. We do not know of any independent observation that can be used to validate the obtained NCE flux at such high spatial and temporal scale. An exception may be inversions and the regional aggregates obtained in the RECCAP synthesis, and we compare our estimates to RECCAP in the manuscript. We have now also included a similar comparison for the ocean regions (section 3.3.2). This comparison shows that the estimates from the synthesis and RECCAP agree well in all regions except the Southern Ocean. At finer spatial and temporal scale, and in some regions, especially the tropics and northern high latitudes, independent trustworthy references are lacking.*

6. In section 3.2, it says that "13% of our runs we obtain a global C source that is consistent with the atmospheric growth rate", what are the spatial distributions of the fluxes from these 13% runs?

*In the original version, 13% of the runs were consistent with the atmospheric growth rate only when assuming neutral exchange in tropical forests. In the revised version we use directly upscaled NEP and none of the runs is consistent with the atmospheric growth rate. This is related to the fact that the uncertainty related to flux separation (dividing NEP into TER and GPP) is not included anymore, which is more consistent because NEP is measured directly by eddy-covariance towers. In addition the large amount of missing fluxes may also inhibit such a constraining exercise, since the magnitude of these omitted fluxes would largely determine which runs would get selected.*

7. L26 (P14): what is the distribution of the different age classes of forests in FLUXNET? Is there solid evidence showing that the year and regrowing forests are overrepresented in FLUXNET?

*So far, this is only a hypothesis and it has not been shown. This hypothesis is a strong candidate in explaining the overestimation of the carbon sink in the tropics. Future research has to demonstrate whether these hypotheses are valid. We have made this clearer in the revised version by writing "Given the difference between NCE and inversions in the tropics (Figure 4), we can assume that a bias of FLUXCOM NEP towards a too high $CO_2$ sink is the main reason why the C budget is not closed in our approach. This raises the question why upscaled NEP has such a strong systematic bias towards a sink, particularly in the tropics (see also Jung et al., 2011). We suspect that the eddy-covariance towers collected in FLUXNET, which provide the empirical basis for the global data driven estimates (see Sect. 2.3.1) do not represent the different age classes of forests very well. For instance, young and regrowing forests with a generally higher-than average NEP are possibly overrepresented in FLUXNET. However, such an age-dependency (Amiro et al., 2010; Coursolle et al., 2012; Hyvönen et al., 2007; Magnani et al., 2007) has not yet been included in global upscaling of NEP. This hypothesis should be tested in future upscaling exercises."*

8. Section 3.5 discusses seasonal cycle and monthly variability. It would be helpful to put this discussion in perspective, e.g., comparing to other independent estimations, so that the readers would know the credibility of this result. It is not clear what are the latitude ranges for the NH and SH in Figure 6.

*The ranges are 90 S-0 for SH and 0-90 N for NH. We have now included a comparison with inversions for the seasonal cycle, which are the only independent estimates based on observational data (section 3.5). As annual variability is already compared to inversions demonstrating large discrepancies (Figure 7) we refrain from comparing monthly variability.*

9. Line 6 (p15), what is the basis for the 50% uncertainty?

*Here we refer to a recent paper on global fire emissions by van der Werf et al, currently in discussion for Earth System Science Data (van der Werf et al., 2017). Estimating uncertainties in fire emission estimates is notoriously difficult. Assuming 50% uncertainty for estimated fire emissions is a best guess assessment, and better quantifying this uncertainty requires an assessment of the burned area estimates as well as new field data on fuel consumption and emission factors. We have rephrased this section as "While GFED4 burned area estimates come with regional uncertainty estimates (Giglio et al., 2013), the actual uncertainty of C emissions from fires are probably much larger, on the order of 50% (van der Werf et al., 2017). The uncertainties of fire emission estimates depend regionally and temporally on the various input data sets such as burned area, small fire burned area, biomass loadings, and combustion completeness. Better quantifying this uncertainty requires an assessment of the burned area estimates as well as new field data on fuel consumption and emission factors. In this study we cannot propagate this uncertainty into the NCE estimates as this would require spatiotemporal error covariance matrices."*

Minor comments
1. In the abstract, "would require an offsetting surface C source of 4.27 +- 0.10 PgC/yr", should the offset be 4.27 + 6.07 PgC/yr in order to have 4.27 PgC atmospheric CO2 growth rate?

*Yes, that is correct. We have reformulated this as "Overall, and because of the over-estimated $CO_2$*

*uptake in tropical lands, our global bottom-up NCE amounts to a net sink of -5.4±2.0 PgC/yr. By contrast, the accurately measured mean atmospheric growth rate of $CO_2$ over 2001-2010 indicates that the true value of NCE is a net $CO_2$ source of 4.3±0.1 PgC/yr. This mismatch of nearly 10 PgC/yr highlights observational gaps and limitations of data-driven models in tropical lands, but also in North America." (The numbers have slightly changed in the revision because we now use directly upscaled NEP instead of GPP-TER).*

2. Line 13-16 (p3), it is not clear what the "background CO2 fluxes" means.

*With background fluxes we mean the fluxes before human disturbance (i.e., before the large increase in fossil fuel emissions). Those are not included in the estimates of the Global Carbon Project, which only discusses the human perturbation. In the revised version, we have avoided the term to avoid confusion and just state "Unlike the GCP global budget of anthropogenic $CO_2$, we consider here the full contemporary exchange of surface-atmosphere $CO_2$ fluxes"*

3. Line 23 (p4), "goal of this study the" should be "goal of this study to"

*Thank you.*

4. Line 16 (p6): what is "schused"?

*Should be "used", has been changed.*

5. Line 21 (p14): What does the "relevant drivers" refer to? Be more specific.

*Additional drivers relevant for upscaling NEP could be, for instance, the disturbance history (e.g. time since the last disturbance) or, closely related, forest age. This is mentioned a few lines higher up. Furthermore soil moisture estimates and information on management practices would help. We now use 'predictors' instead of 'drivers' and have rephrased the sentence as "However, not all of the relevant predictors (i.e. disturbance maps, management practices, soil moisture) are currently available to be included in empirical upscaling exercises (Tramontana et al., 2016)."*

6. Line 29 (p14): what does the "global driver" refer to?

*By this we refer to the fact that there is no global map of forest age, which could be used as an additional driver for upscaling NEP (see also comment above). We have omitted the part with the "global driver" and reformulated this section. We also use the term 'predictor' instead of 'driver', which is more intuitive. See reply to major comment 7 for the complete text.*

*Additional References:*

[revised manuscript text omitted]

---

## Author Response (AR2)

*Point-to-point responses to the reviewer's comments are presented below in italic.*

Reviewer #2 comments:

This paper has improved from its original version, but still requires substantial revision. The methods are unclear with insufficient detail. Comparisons to other work are provided, but these other approaches are also poorly described so the readers cannot actually understand what the comparisons indicate. Uncertainty is inconsistently presented. Handwaving statements about independence of methods are, in fact, untrue in the case of the ocean. This does not provide the reader confidence in the analysis.

*We have revised the section explaining the uncertainty assessment. We have now also more carefully highlighted which estimates are truly independent, and where some overlap exists between the different datasets.*

The paper does not lead to a substantial enhancement in scientific understanding, but reads instead like a technical report on a data product. Because of this, I think the paper would belong better in Earth System Science Data than Biogeosciences.

Comments:

Page 3, line 24 – The GCP separation of % to land and ocean needs to be qualified with the timeframe to which it refers. This is not a steady state partitioning as suggested by the sentence as written.

*We have adjusted this statement to the years 2000-2009 (the time period closest to the time period discussed in the paper). The slightly adjusted numbers (45%, 27% and 27% respectively) are taken from Table 8 in Le Quéré et al. (2015).*

Page 4, line 11 - The "full contemporary" exchange would include the large, persistent ocean outgassing in tropics, and uptake at high latitudes See Figure 6.1 of IPCC AR5 ch6, Ciais et al. 2013, these fluxes are +-80 PgC/yr. Similarly "Full contemporary fluxes" for the land would be about +-120PgC/yr as in the original version of this MS. So this net flux is not the "full contemporary exchange", but only the net contemporary flux. The authors need to explain these differences and find appropriate language that clarifies for the reader.

*As discussed in the response to the reviewers' comments, we focus here on the net CO2 exchange between the surface and the atmosphere. Further, there are no bottom-up spatiotemporal explicit gross fluxes available for the ocean-atmosphere exchange. To make this clear, we have adjusted this sentence to "we consider here the full contemporary net exchange of surface-atmosphere $CO_2$ fluxes".*

Section 2.1 - The reader cannot understand to what "ensemble members" are being referred when these have not yet been introduced. The "random draw" approach is not clear. Where do the numbers come from in 10x10x…. ??? 2 is presumably the ocean, but this placement in the equation on line 24 does not correspond to equation 1, enhancing confusion. Figure 1 is not understandable. Table 1 could perhaps help, but there are more than 9 "estimates" there.
This presentation needs much work so that the reader can fully understand the approach.

*We apologize, there was an additional "10" in the equation that was not needed. The correct number of possible estimates is 10x50x8x10x2=80000 (Marine x Rivers x NEP x Crops x $E_{LUC}$, for the 4*

*remaining variables we only have 1 estimate, see Table 1). We revised the beginning of this section to explain the random-draw approach better. "For all 9 variables contributing to NCE of equation 1, we obtained multiple spatiotemporally explicit estimates directly from the raw data products. These estimates are expected to sample the uncertainty in each variable. We could obtain 10 different estimates for the terms Marine and Crops respectively, 50 estimates for Rivers, 8 estimates for NEP, 2 estimates for ELUC, and only 1 estimate for each of the remaining 4 variables (Table 1). To estimate NCE including spatiotemporally explicit uncertainties, we randomly draw ensemble members from each of these 9 terms to create an integrated for NCE (Eq. 1). With the available estimates, we could in principle create 10x50x8x10x2 = 80000 spatiotemporal explicit estimates of NCE. From these 80000 possible NCE estimates we randomly select 200 (to reduce computational expenses) to construct the NCE ensemble, which is used in the remainder of the paper."*

Section 2.2.4 – State here how many final estimates are produced (2 for marine) so that the reader can understand the error propagation. Do the same for the other sections.

*Thank you for this suggestion. We have done this for all variables.*

Section 2.5 - Yearly variability in precisely which of the fluxes above? The authors should not assume the reader knows what an "inversion" would provide. Also, the Median was used in RECCAP - what justifies the use of the Mean here?

*We aim for a comparison of yearly variability, spatial patterns and latitudinal bands of NCE. We have made that explicit in the first sentence of 2.5. We have also now introduced atmospheric inversion by adding the sentences "Atmospheric $CO_2$ inversions estimate the spatiotemporal explicit $CO_2$ exchange between the Earth surface and atmosphere using atmospheric $CO_2$ measurements and a transport model. Inversions have a closed budget by construction."*

*In Figures 4, 7, and 8 we now present the median and the interquartile range to be consistent with the RECCAP comparisons.*

Page 10, line 30 – Support this statement about model tuning with evidence from the literature.

*We refer to Schwalm et al. (2015). Models may be tuned to the few observational constraints available (e.g., CO2 growth rate), making it difficult to uncover compensatory errors. In contrast, bottom-up approaches do not have such higher-level observational constraints.*

Page 12, line 18 – Define "bookkeeping" – not all readers know this lingo

*We have added the following explanation for bookkeeping: "A bookkeeping model combines local or regional observation-based estimates of C stocks before land-use change and trajectories of C stocks after land use change including slash, biomass, soil carbon, and harvested wood products with changing areas of different land use types (Houghton and Nassikas, 2017)."*

Page 12, line 23-27 – This section on comparison to Ciais et al (in revision) is unintelligible because the reader does not know what this other paper does and so what does this comparison actually mean? Later it is stated that Ciais uses an "independent" method, but the authors have not described what this method is sufficiently.

*We added the following sentence to explain the approach of Ciais et al.: "Ciais et al. (in revision) compiled regional estimates of land-atmosphere $CO_2$ fluxes in the RECCAP regions from original*

*publications, completed by river CO$_2$ outgassing fluxes from Raymond et al. (2013) and Lauerwald et al. (2015). Thus, these estimates are not fully independent from those presented in this study because they use the same river fluxes, fire emissions, and FF emission. All other fluxes are independent."*

Page 13., line 14. – It is untrue that the RECCAP ocean estimates are independent of those used here. Much of the same pCO2 data is found in the Takahashi flux estimate, a major basis for RECCAP estimates, and that in SOCAT on which these estimates are based. While the Takahashi flux estimate is not directly used here, it is not correct that the ones used are fully independent. The authors should not overstate independence.

*We agree. The independence is more in the methods that estimate CO$_2$ exchange over oceans. We have made this clearer by stating "While there is large overlap between the pCO$_2$ data used in the RECCAP estimates and our NCE estimates over oceans, different independent methods have been used to obtain flux estimates of ocean CO$_2$ exchange."*

Page 13, line 29. This comparison does not "suggest" less IAV in this result than in inversion, it directly shows it. Please be direct in wording.

*We have replaced "suggest" with "shows".*

Figure 2 – Uncertainties must be included on this figure.

*Uncertainties have been added.*

References:

[revised manuscript text omitted]